# SIKs control osteocyte responses to parathyroid hormone

Marc N. Wein[1], Yanke Liang[2], Olga Goransson[3], Thomas B. Sundberg[4], Jinhua Wang[2], Elizabeth A. Williams[1], Maureen J. O'Meara[1], Nicolas Govea[1], Belinda Beqo[1], Shigeki Nishimori[1], Kenichi Nagano[5], Daniel J. Brooks[1,6], Janaina S. Martins[1], Braden Corbin[1], Anthony Anselmo[7], Ruslan Sadreyev[7], Joy Y. Wu[8], Kei Sakamoto[9,†], Marc Foretz[10], Ramnik J. Xavier[11,12,13], Roland Baron[1,5], Mary L. Bouxsein[1,6], Thomas J. Gardella[1], Paola Divieti-Pajevic[14], Nathanael S. Gray[2] & Henry M. Kronenberg[1]

Parathyroid hormone (PTH) activates receptors on osteocytes to orchestrate bone formation and resorption. Here we show that PTH inhibition of *SOST* (sclerostin), a WNT antagonist, requires HDAC4 and HDAC5, whereas PTH stimulation of RANKL, a stimulator of bone resorption, requires CRTC2. Salt inducible kinases (SIKs) control subcellular localization of HDAC4/5 and CRTC2. PTH regulates both HDAC4/5 and CRTC2 localization via phosphorylation and inhibition of SIK2. Like PTH, new small molecule SIK inhibitors cause decreased phosphorylation and increased nuclear translocation of HDAC4/5 and CRTC2. SIK inhibition mimics many of the effects of PTH in osteocytes as assessed by RNA-seq in cultured osteocytes and following *in vivo* administration. Once daily treatment with the small molecule SIK inhibitor YKL-05-099 increases bone formation and bone mass. Therefore, a major arm of PTH signalling in osteocytes involves SIK inhibition, and small molecule SIK inhibitors may be applied therapeutically to mimic skeletal effects of PTH.

[1] Endocrine Unit, Department of Medicine, Massachusetts General Hospital, Harvard Medical School, 50 Blossom Street, Boston, Massachusetts 02114, USA. [2] Dana Farber Cancer Institute, Department of Biologic Chemistry and Molecular Pharmacology, Harvard Medical School, 450 Brookline Avenue, Boston, Massachusetts 02215, USA. [3] Department of Experimental Medical Sciences, Lund University, Box 188, SE-221 00 Lund, Sweden. [4] Center for the Development of Therapeutics, Broad Institute, 415 Main Street, Cambridge, Massachusetts 02142, USA. [5] Harvard School of Dental Medicine, Department of Oral Medicine, Infection, and Immunity, 188 Longwood Avenue, Boston, Massachusetts 02115, US. [6] Center for Advanced Orthopaedic Studies, Department of Orthopedic Surgery, Beth Israel Deaconess Medical Center, 330 Brookline Avenue, Boston, Massachusetts 02215, USA. [7] Department of Molecular Biology, Massachusetts General Hospital, Harvard Medical School, 185 Cambridge Street, Boston, Massachusetts 02114, USA. [8] Division of Endocrinology, Department of Medicine, Stanford University School of Medicine, 300 Pasteur Dr a175, Stanford, California 94305, USA. [9] MRC Protein Phosphorylation and Ubiquitylation Unit, College of Life Sciences, University of Dundee, Dundee DD1 5EH, Scotland, UK. [10] INSERM U1016, Institut Cochin, CNRS UMR8104, Universite Paris Descartes Sorbonne Pairs Cite, Paris 75013, France. [11] Gastrointestinal Unit and Center for the Study of Inflammatory Bowel Disease, Department of Medicine, Massachusetts General Hospital, 55 Fruit Street, Boston, Massachusetts 02114, USA. [12] Center for Computational and Integrative Biology, Massachusetts General Hospital, Harvard Medical School, 55 Fruit Street, Boston, Massachusetts 02114, USA. [13] Program in Medical and Population Genetics, Broad Institute, 415 Main Street, Cambridge, Massachusetts 02142, USA. [14] Henry M. Goldman School of Dental Medicine, Boston University, 100 E Newton Street, Boston, Massachusetts 02118, USA. † Present address: Nestlé Institute of Health Sciences SA, Campus EPFL, Quartier de l'innovation, Bâtiment G, 1015 Lausanne, Switzerland. Correspondence and requests for materials should be addressed to H.M.K. (email: kronenberg@helix.mgh.harvard.edu).

Osteoporosis is a serious problem in our ageing population, with fragility fractures costing $25 billion annually[1]. Novel treatments are needed to boost bone mass. Osteocytes, cells buried within bone, orchestrate bone remodelling by secreting endocrine and paracrine factors[2]. Central amongst these are RANKL (encoded by the *TNFSF11* gene), the major osteocyte-derived osteoclastogenic cytokine[3,4] and an FDA-approved osteoporosis drug target, and sclerostin (encoded by the *SOST* gene), an osteocyte-derived WNT pathway inhibitor that blocks bone formation by osteoblasts[5] and current osteoporosis drug target[6].

When given once daily, parathyroid hormone (PTH), is the only approved osteoporosis treatment agent that stimulates new bone formation. The proximal signalling events downstream of Gsα-coupled PTH receptor signalling in bone cells are well-characterized[7], but how cyclic adenosine monophosphate (cAMP) generation in osteocytes is linked to gene expression changes remains unknown. *SOST* and *RANKL* are well-established target genes important for the physiological effects of PTH on osteocytes. Among the mechanisms through which PTH stimulates new bone formation, down-regulation of *SOST* expression in osteocytes plays an important role[8–10]. PTH also stimulates bone catabolism, in large part through stimulation of osteoclastogenesis via inducing *RANKL*[11–14], which may limit its therapeutic efficacy.

We have previously described a role for the class IIa histone deacetylase HDAC5 as a negative regulator of MEF2C-driven *SOST* expression, both *in vitro* in Ocy454 osteocytic cells[15] and *in vivo*[16]. Class IIa HDACs are uniquely endowed with N-terminal extensions that allow them to sense and transduce signalling information[17]. When phosphorylated, class IIa HDACs are sequestered in the cytoplasm via binding to 14-3-3 proteins. When de-phosphorylated, they are able to translocate to the nucleus to inhibit MEF2-driven gene expression[18]. Like class IIa HDACs, cAMP-regulated transcriptional coactivators (CRTC) proteins shuttle from the cytoplasm to the nucleus where they function as CREB coactivators[19]. Both HDAC4/5 and CRTC2 are known substrates of salt inducible kinases (SIKs)[19–22], and *SIK3* deficiency in growth plate chondrocytes increases nuclear HDAC4 and delays MEF2-driven chondrocyte hypertrophy[21].

Here, we show that PTH signalling in osteocytes uses both HDAC5 and the closely related family member HDAC4 to block MEF2C-driven *SOST* expression. In addition, PTH-stimulated *RANKL* expression requires CRTC2. PTH signalling, via cAMP, inhibits SIK2 cellular activity in osteocytes. SIK inhibition, both *in vitro* and *in vivo*, achieved via the small molecule YKL-05-093, is sufficient to mimic many of the effects of PTH, including lower levels of HDAC4/5/CRTC2 phosphorylation, *SOST* inhibition and *RANKL* stimulation. Strikingly, a major arm of PTH signalling in osteocytes involves SIK inhibition, as revealed by RNA-seq analysis of PTH- versus YKL-05-093-treated osteocytes. Finally, we demonstrate that YKL-05-099 (ref. 23), an analogue of YKL-05-093 with properties making it suitable for targeting SIKs *in vivo*, is able to boost osteoblast numbers, bone formation, and bone mass in mice. In summary, our results demonstrate that a PTH receptor/cAMP/SIK/class IIa HDAC/CRTC axis has a crucial role in osteocyte biology.

## Results

**Class IIa HDACs control bone mass through *SOST*.** Having previously demonstrated that HDAC5 blocks MEF2C-driven *SOST* expression in osteocytes[16], we sought to determine whether *HDAC5* and *SOST* interact *in vivo* to control bone mass. Two complementary approaches demonstrated that this was the case. First, compound heterozygosity of *HDAC5* and *SOST* rescued the cortical and trabecular high bone mass phenotype of *SOST*[+/−]

mice (Supplementary Fig. 1A–C). Second, anti-sclerostin antibody treatment rescued the trabecular osteopenia present in *HDAC5*[−/−] animals (Supplementary Fig. 1D), which have high levels of *SOST* expression[16].

With evidence that HDAC5 control of *SOST* is physiologically important, we asked if other class IIa HDACs function in osteocytes. We[16] and others[24] have previously reported that *HDAC5*[−/−] mice display mild trabecular osteopenia. For these studies, we extended our analyses to include the closely related family member *HDAC4* for two reasons. First, endogenous MEF2C immunoprecipitates from Ocy454 cells contained HDAC4 in addition to HDAC5 (Fig. 1a and ref. 16). Second, while no obvious skeletal phenotype was observed when *HDAC4* was deleted from osteocytes using DMP1-Cre[25], compound deletion of both *HDAC4* and *HDAC5* led to a skeletal phenotype not observed in either single mutant strain, characterized by severe trabecular osteopenia (Supplementary Table 1 and Supplementary Fig. 1F for results of static and dynamic histomorphometry results), increased osteocyte density (Fig. 1b,c), disorganized, 'woven' cortical bone (Fig. 1d), failure to respond to sclerostin antibody (Supplementary Fig. 1D), and reduced endocortical bone formation (Supplementary Fig. 1E). As we previously reported, mice lacking *HDAC5* alone show mild cancellous osteopenia and reduced markers of bone formation by histomorphometry[16].

**PTH signals through HDAC4 and HDAC5 to suppress *SOST*.** We next asked whether PTH, a known suppressor of *SOST* expression[8], worked through HDAC4, HDAC5, or both. PTH treatment of Ocy454 cells caused translocation from the cytosol to the nucleus of both HDAC4 and HDAC5 (Fig. 2a). When phosphorylated, class IIa HDACs are predominantly cytoplasmic through retention by 14-3-3 proteins[17]. When dephosphorylated, class IIa HDACs translocate to the nucleus where they potently inhibit MEF2-driven gene expression in muscle[26,27]. In neurons, HDAC5 nuclear import is additionally inhibited by *CDK5*-mediated phosphorylation at S279 (ref. 28). PTH signalling reduced phosphorylation of HDAC4 at S246/S632 and, to a lesser extent, HDAC5 at S259/S279 (Fig. 2b, Supplementary Fig. 2A). Others have over-expressed *HDAC5* in a rat osteosarcoma cell line to demonstrate that mutation of these serines to alanine led to PTH-independent nuclear import[29]. PTH-induced loss of phosphorylation and nuclear translocation of HDAC4/5 requires cAMP signalling, as evidenced by the fact that these events did not occur in cells lacking Gsα via CRISPR/Cas9-mediated genome editing (Fig. 2c,d, and Supplementary Fig. 2B–E). As previously described[15,30,31], Gsα deficiency significantly increases sclerostin production by osteocytes. However, reducing MEF2C levels via shRNA or by over-expressing a constitutively nuclear super-repressor form of HDAC5 rescued this phenotype (Supplementary Fig. 2F–I), consistent with the model that Gsα deficiency increases sclerostin production via a gain-of-function MEF2C phenotype.

To determine the roles of HDAC4/5 in mediating PTH actions, we generated osteocytes lacking HDAC4 (via CRISPR/Cas9-mediated deletion, Supplementary Fig. 3A–E), HDAC5 (via lentiviral-mediated shRNA) or both (Fig. 2e). While cells lacking HDAC4 or HDAC5 alone showed normal suppression of *SOST* expression in response to PTH, deletion of both HDAC4 and HDAC5 abolished PTH-induced *SOST* down-regulation (Fig. 2f, left). Importantly, HDAC4/5-deficient cells showed preserved PTH-induced *RANKL* up-regulation (Fig. 2f, right). Chromatin IP revealed that PTH signalling reduces MEF2C binding to the +45 kB downstream *SOST* enhancer (Fig. 2g); this occurs rapidly, at time points before observed reductions in

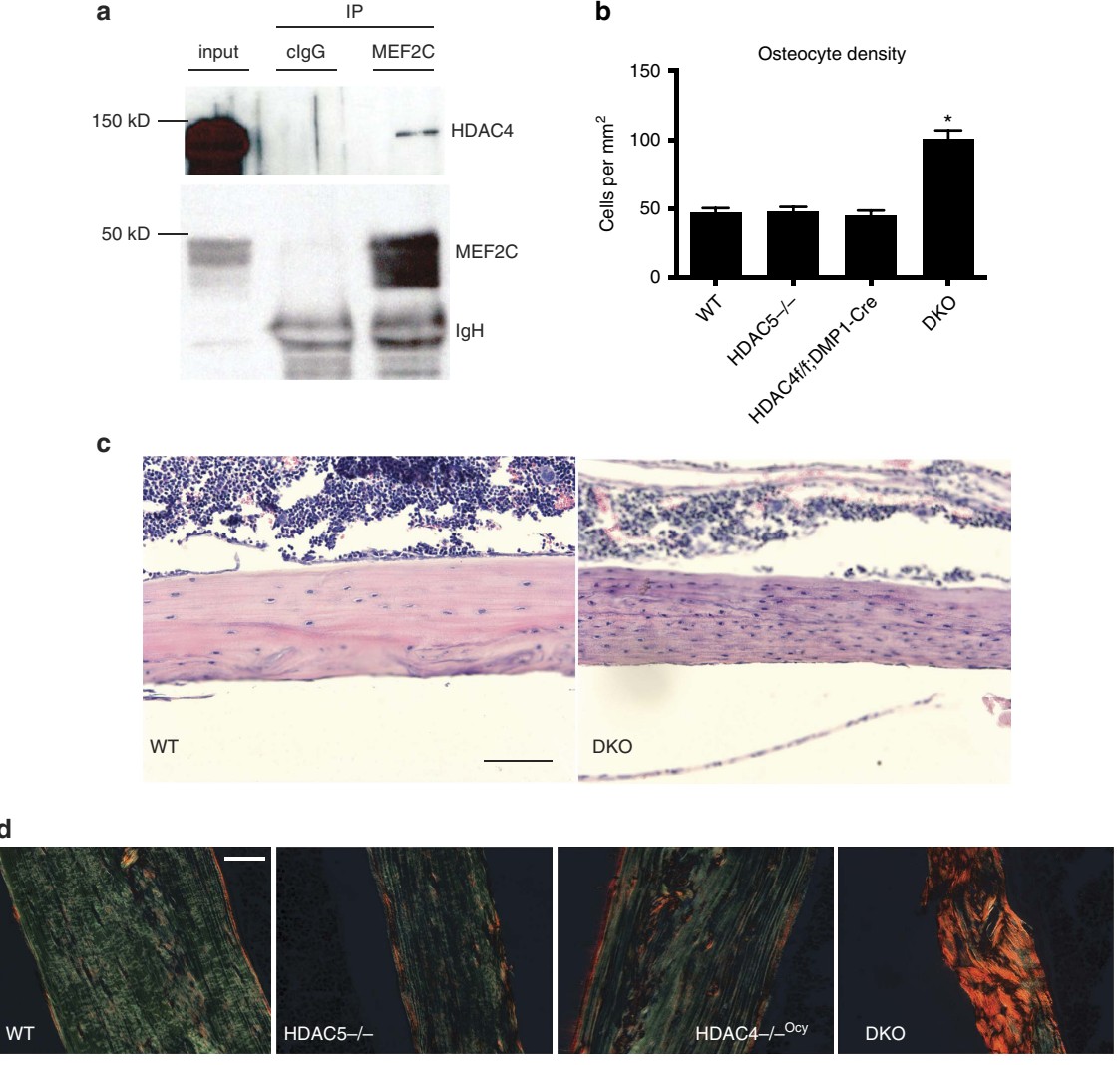

**Figure 1 | *HDAC4* and *HDAC5* control osteocyte biology *in vivo*. (a)** Endogenous MEF2C was immunoprecipitated from Ocy454 cells, followed by immunoblotting for the indicated proteins. Data shown are representative of $n = 3$ independent experiments. **(b)** Osteocyte density in cortical bone 3 mm below the growth plate. 4–5, 8 week old male mice per genotype was analysed, *indicates $P < 0.01$ versus WT by student's unpaired two tailed *t*-test. **(c)** Representative H + E section demonstrating increased osteocyte density and disorganized cortical bone in DKO (*HDAC4f/f; HDAC5$^{-/-}$;DMP1-Cre*) mice. Scale bar, 40 μm. **(d)** Sections were stained with Sirius Red and analysed under polarized light to view collagen fibre organization. Disorganized collagen fibers are only seen in DKO sections. Scale bar, 40 μm. Error bars indicate s.e.m for all figures.

*MEF2C* mRNA levels (Supplementary Fig. 3F and refs 32,33). HDAC4/5-deficient cells showed increased MEF2C binding at baseline, and failed to reduce MEF2C *SOST* enhancer occupancy in response to PTH (Fig. 2h).

To determine the relevance of *HDAC4/5* in mediating PTH actions *in vivo*, *HDAC4/5*-deficient mice were treated with PTH, and acute effects were measured 90 min later. While bone *RANKL* levels increased comparably across all four genotypes (*WT*, *HDAC5$^{-/-}$*, *HDAC4f/f;DMP1-Cre*, and *HDAC4f/f;HDAC5$^{-/-}$; DMP1-Cre*), *HDAC4/5*-deficient mice were unique in that *SOST* levels failed to decrease following PTH treatment (Fig. 3a,b). At the protein level, PTH administration significantly decreased the numbers of sclerostin-immunoreactive cortical osteocytes in all genotypes tested except in *HDAC4/5*-deficient animals (Fig. 3c,d). Taken together, these results indicate that HDAC4 and HDAC5 are downstream of PTH receptor signalling, and are required for PTH-mediated *SOST* suppression, both *in vitro* and *in vivo*.

While *SOST* is a well-studied PTH target genes, it represents a small portion of the transcriptome regulated by parathyroid hormone (see below). Underscoring this point, once daily

intermittent PTH treatment leads to comparable gains in trabecular bone density in mice lacking *HDAC4* in osteocytes, *HDAC5* or both (Supplementary Fig. 4A). Therefore, although class II HDACs are required for acute PTH-induced changes in *SOST* expression, other signalling arms and target genes downstream of the PTH receptor must exist that are important for the pharmacologic effects of this hormone.

**SIK2 is inhibited by PTH and required for PTH signalling.** We next addressed the signalling mechanisms used between activation of the PTH receptor and decreased phosphorylation of HDAC4/5. In chondrocytes *in vitro*, PTHrP drives HDAC4 into the nucleus via PP2A-mediated dephosphorylation, which can be blocked by okadaic acid[34]. Surprisingly, okadaic acid did not block PTH-mediated decreased *HDAC4/5* phosphorylation or *SOST* suppression in Ocy454 cells (Supplementary Fig. 4B,C). Similarly, PTH-induced decreases in HDAC4/5 phosphorylation and *SOST* suppression were intact when PP2A catalytic subunit levels were reduced via shRNA (Supplementary Fig. 4D,E).

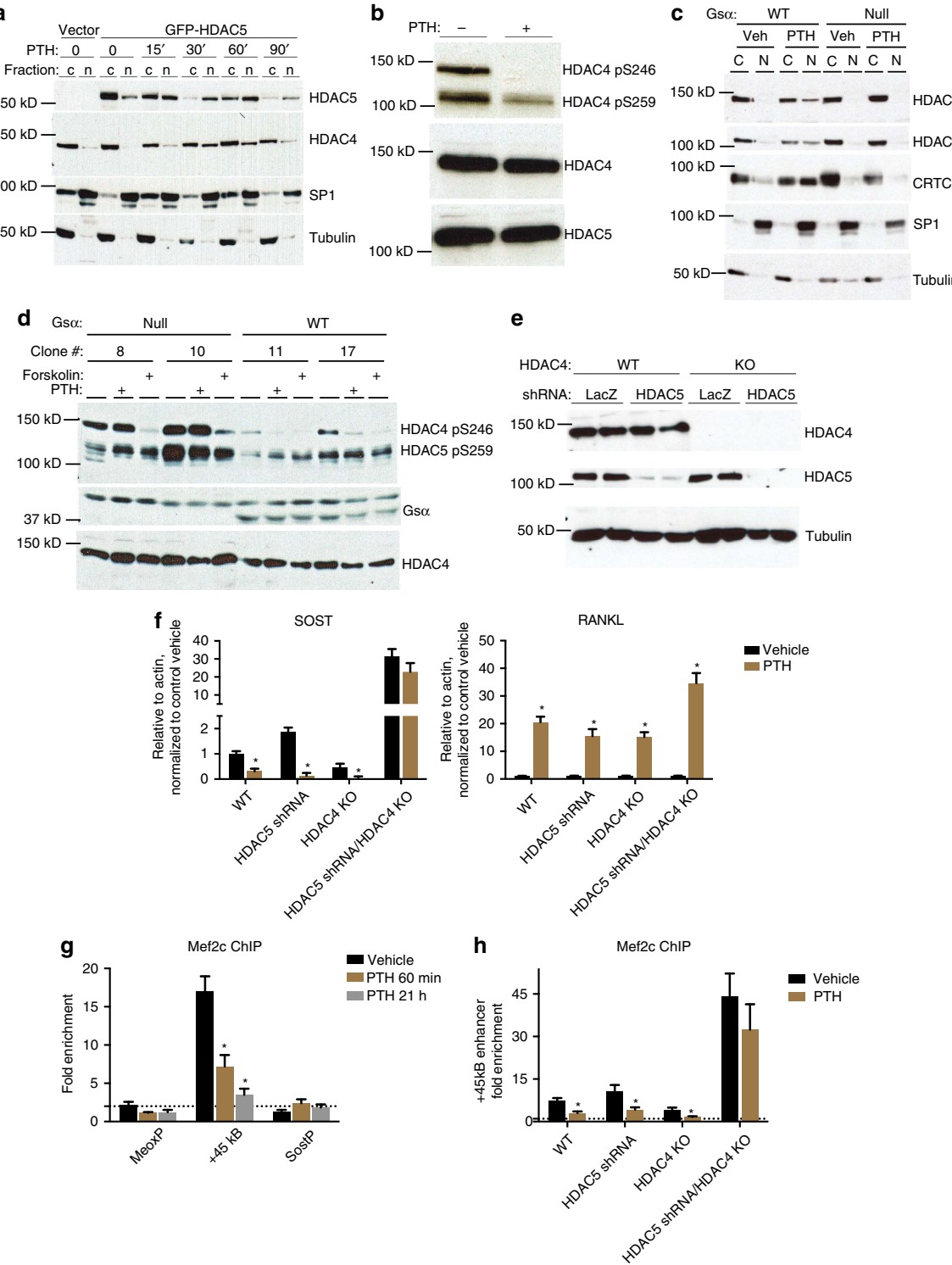

**Figure 2 | Class IIa HDACs are required for PTH-induced *SOST* suppression *in vitro*.** (**a**) Ocy454 cells were transfected with GFP-HDAC5 and then treated with PTH (50 nM) for the indicated times. Cytosolic (c) and nuclear (n) lysates were prepared and immunoblotted as indicated. (**b**) Ocy454 cells were treated with PTH (50 nM) for 30 min. Whole cell lysates were prepared and immunoblotted as indicated. Similar results were observed in four independent experiments. (**c**) Ocy454 cells with (WT, clone 17) and without (Null, clone 8) Gsα were treated with PTH (50 nM for 30 min) and analysed as in (**a**). (**d**) Ocy454 cells with (WT) and without (Null) Gsα were treated with either PTH (50 nM) or forskolin (5 μg ml⁻¹) for 30 min and analysed as in **b**. (**e**) Ocy454 cells were exposed to the indicated combinations of *HDAC4*-targeting sgRNAs (with Cas9) and *HDAC5* shRNA-expressing lentiviruses, and whole cell lysates were analysed by immunoblotting as indicated. (**f**) WT, *HDAC5* shRNA, *HDAC4* KO and DKO Ocy454 cells were treated with PTH (1 nM) for 4 h, and *SOST* (left) and *RANKL* (right) mRNA transcript abundance was measured by RT-qPCR. For all cell culture experiments therein, values represent mean of $n = 3$ biologic replicates. *indicates $P < 0.05$ comparing the effects of PTH to vehicle for each cell line. Error bars represent SEM. (**g,h**) MEF2C chromatin immunoprecipitation was performed, and enrichment for the +45 kB enhancer determined (relative to control IgG ChIP). *indicates $P < 0.05$ comparing fold enrichment of PTH versus vehicle by student's unpaired two tailed *t*-test.

Okadaic acid and PP2A shRNA efficacy was confirmed in these experiments based on observed increases in HDAC4 S246 phosphorylation (Supplementary Fig. 4B,D). Taken together, these results suggest that, unlike in chondrocytes, in osteocytes PTH-stimulated decreased phosphorylation of HDAC4/5 is not mediated by activation of PP2A.

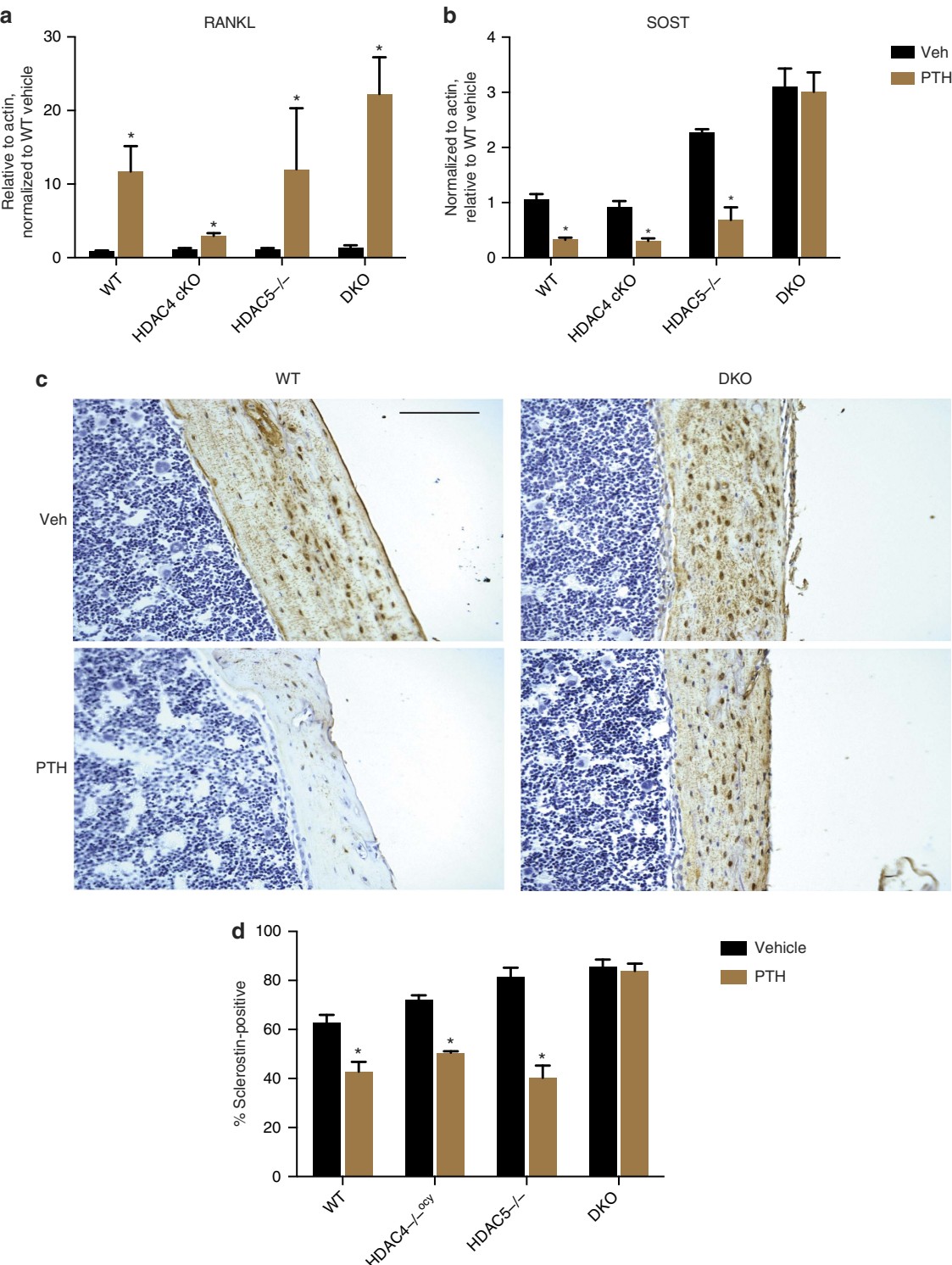

**Figure 3 | Class IIa HDACs are required for PTH-induced *SOST* suppression *in vivo*.** (**a**,**b**) 6 week old male mice of the indicated genotype were treated with vehicle or PTH (1-34, 300 μg kg$^{-1}$) and euthanized 90 min later. Bone RNA was obtained and *RANKL* and *SOST* transcript abundance was determined by RT-qPCR. * indicates $P < 0.05$ comparing vehicle and PTH for each genotype. $N =$ at least 6 mice per group were analysed. (**c**) Representative photomicrographs of sclerostin immunohistochemistry from WT and DKO mice treated with vehicle or PTH. Scale bar, 40 μm. (**d**) Quantification of immunohistochemistry results. Cortical osteocytes in a fixed region of bone 3 mm below the tibial growth plate were counted and scored as either sclerostin-positive or negative. $N = 6$ mice per group were analyzed. * indicates $P < 0.01$ comparing vehicle and PTH for each genotype by student's unpaired two tailed *t*-test.

To explore candidate kinases whose activity might mediate the actions of PTH on HDAC4/5, we examined salt inducible kinases (SIKs), AMPK family members reported to function as class IIa HDAC N-terminal kinases[20,35]. Subcellular fractionation experiments revealed that both SIK2 and SIK3 proteins are predominantly cytoplasmic in osteocytes (Supplementary Figure 4F). Combined silencing of both *SIK2* and *SIK3* in Ocy454 cells significantly decreased HDAC4/5 N-terminal phosphorylation (Fig. 4a).

cAMP signalling in adipocytes and hepatocytes inhibits SIK2 activity via protein kinase A (PKA)-mediated phosphorylation, which in turn sequesters SIK2 from its substrates[36–38]. PTH signalling in osteocytes triggered SIK2 phosphorylation at S343, S358 and T484 (Fig. 4b). PKA-mediated SIK3 phosphorylation was not triggered by PTH signalling (Fig. 4b). Notably, PTH-stimulated SIK2 S358 phosphorylation occurred rapidly, faster than the fall in HDAC4/5 phosphorylation levels (Fig. 4c). Importantly, *SIK2*-silenced cells showed normal up-regulation of the PTH target gene *CITED1* (ref. 39) (Fig. 4d). In contrast, PTH-induced decreases in HDAC4/5 phosphorylation (Fig. 4e) and *SOST* suppression (Fig. 4f) did not occur in SIK2-silenced cells. Interestingly, PTH-induced *RANKL* upregulation, an HDAC4/5-independent phenomenon (Figs 2f and 3a) also did not occur in SIK2-deficient osteocytes (Fig. 4g), suggesting that another SIK substrate is involved in PTH-mediated RANKL gene induction. SIK3-deficient cells showed normal PTH responses (Fig. 4d,f,g), as predicted by the fact that this protein is not phosphorylated in response to PTH signalling. cAMP responses to PTH were blunted in SIK2-silenced Ocy454 cells but were clearly present at PTH levels above 4 nM (Supplementary Fig. 4G). Nevertheless, this effect on cAMP levels in response to PTH is unlikely to explain the effects of *SIK2* silencing. Forskolin-induced cAMP up-regulation was normal in SIK2-deficient cells, yet this agent failed to regulate *SOST* or *RANKL* expression in the absence of SIK2 (Fig. 4h).

To determine the relevance of *SIK2* in mediating PTH actions *in vivo*, mice lacking *SIK2* in *DMP1*-expressing cells (including osteocytes) were treated with PTH, and acute effects were measured in bone 120 min later. Figure 4i shows that DMP1-Cre deletion of *SIK2* led to a significant reduction in *SIK2*, but not PTH receptor, mRNA levels in bone. Similar to the results in Ocy454 cells, PTH-induced *CITED1* up-regulation was preserved in *SIK2*$^{OcyKO}$ mice (Fig. 4j). However, PTH-induced *SOST* and *RANKL* gene regulation did not occur in the absence of SIK2 (Fig. 4k).

*RANKL* is a known PTH target gene; previous studies have suggested an important role for CREB, through binding to an enhancer 75 kB upstream of the transcription start site[12–14,40,41]. While CREB itself is not a known SIK substrate, the CRTC CREB coactivator proteins are[19]. All three CRTC proteins are expressed in osteocytes; therefore, levels of each were reduced individually using shRNA. Only *CRTC2* silencing was sufficient to block PTH-induced *RANKL* up-regulation (Fig. 4l). Supplementary Fig. 4J shows that PTH-induced cAMP generation was normal in CRTC2-deficient cells. PTH promoted CRTC2 nuclear translocation in a Gsα-dependent manner (Fig. 2c), and CRTC2 inducibly associated with the − 75 kB 'D5' *RANKL* enhancer[42] following PTH treatment (Fig. 4m). In summary, these results demonstrate that two key SIK substrates, HDAC4/5 and CRTC2, play major roles in PTH-mediated regulation of *SOST* and *RANKL* expression, respectively.

**SIK inhibitors regulate *SOST* and *RANKL* expression**. Gene ablation studies *in vitro* and *in vivo* suggested that SIK2 is required for PTH to regulate *SOST* and *RANKL* expression, and

that PTH signalling leads to PKA-mediated SIK2 inhibition. Therefore, we wondered whether acute inhibition of SIK kinase activity in otherwise normal cells or mice would be sufficient to mimic these actions of PTH. HG-9-91-01 is a small molecule kinase inhibitor with demonstrated biologic activity against SIKs in cultured macrophages, dendritic cells and hepatocytes[37,38,43,44]. However, HG-9-91-01 is not SIK-specific and is not suitable for *in vivo* use; therefore, we screened for analogues based on the goals of improved specificity and pharmacokinetics. These efforts ultimately led to the identification of YKL-04-114 and its closely related analogue YKL-05-093 (Fig. 5a). The $K_d$ of YKL-05-093 for SIK2 is 7.1 nM, and its activity against a panel of 96 recombinant kinases is shown in Supplementary Table 2 (here SIK refers to SIK1 and QSK refers to SIK3) and shown graphically in Supplementary Fig. 5A. YKL-04-114 or YKL-05-093 treatment of Ocy454 cells led to rapid, dose-dependent decreases in HDAC4/5 phosphorylation (Fig. 5b,c), and increased nuclear translocation of HDAC4 and CRTC2 (Fig. 5d). YKL-04-114 caused rapid and potent *SOST* suppression and *RANKL* up-regulation (Fig. 5e) without increasing cAMP levels (Supplementary Fig. 5B). Optimal efficacy at the level of HDAC4/5 phosphorylation (∼1 μM, Fig. 5c) and gene expression (∼0.5 μM, Fig. 5e) occurred at comparable doses.

Importantly, treatment with YKL-05-093 did not decrease HDAC4 S246 phosphorylation or cause *SOST* suppression in osteocytes lacking SIK2 and SIK3 (Fig. 5f,g). In addition, PTH and YKL-05-093-mediated stimulation of *RANKL* expression was abrogated in cells lacking CRTC2 (Fig. 5h). So although YKL-05-093 does target other kinases *in vitro*, its cellular actions studied here depend on the presence of SIK2 and SIK3.

On the basis of our model that YKL-05-093 functions as a SIK inhibitor downstream of PTH-stimulated cAMP generation, one would predict that the inhibitor would regulate gene expression in Gsα-deficient osteocytes. Indeed, YKL-05-093 treatment of Gsα-deficient Ocy454 caused *SOST* suppression and *RANKL* up-regulation with effects similar to forskolin, except, as expected based on its inability to increase cellular cAMP levels (Supplementary Fig. 5B), YKL-05-093 did not increase SIK2 S358 phosphorylation (Fig. 5i,j).

**Small molecule SIK inhibitors mimic PTH action *in vitro***. The ability of YKL-05-093 to mimic the effects of PTH with respect to *SOST* and *RANKL* gene regulation supports the hypothesis that the actions of YKL-05-093 might mimic the effects of PTH on many genes. We therefore performed RNA-seq on Ocy454 cells treated for four hours with vehicle, PTH (1 nM) or YKL-05-093 (0.5 μM) to determine the overlap in global gene regulation by these two agents. Significantly 446 genes were (>2 fold, FDR < 0.05) regulated by PTH, and 257 genes were significantly regulated by YKL-05-093. Of the 446 PTH-regulated genes, 142 (32%) were co-regulated in the same direction by YKL-05-093 (Fig. 6a,b, Supplementary Table 3 for differentially expressed genes and Supplementary Dataset 1 for all RNA-seq data). This significant overlap was not due to random chance (Supplementary Fig. 6A,B). Gene ontology analysis for the genes regulated by both PTH and YKL-05-093, is shown in Supplementary Fig. S6C,D: many of the co-regulated genes fit into categories of interest such as 'ossification' and 'mesenchyme development'.

Overall, six clusters of differentially-expressed genes were identified: those up-regulated by PTH alone (172 genes), YKL-05-093 alone (56 genes) and both PTH and YKL-05-093 (97 genes), and those down-regulated by PTH alone (132 genes), YKL-05-093 alone (59 genes) and both PTH and YKL-05-093

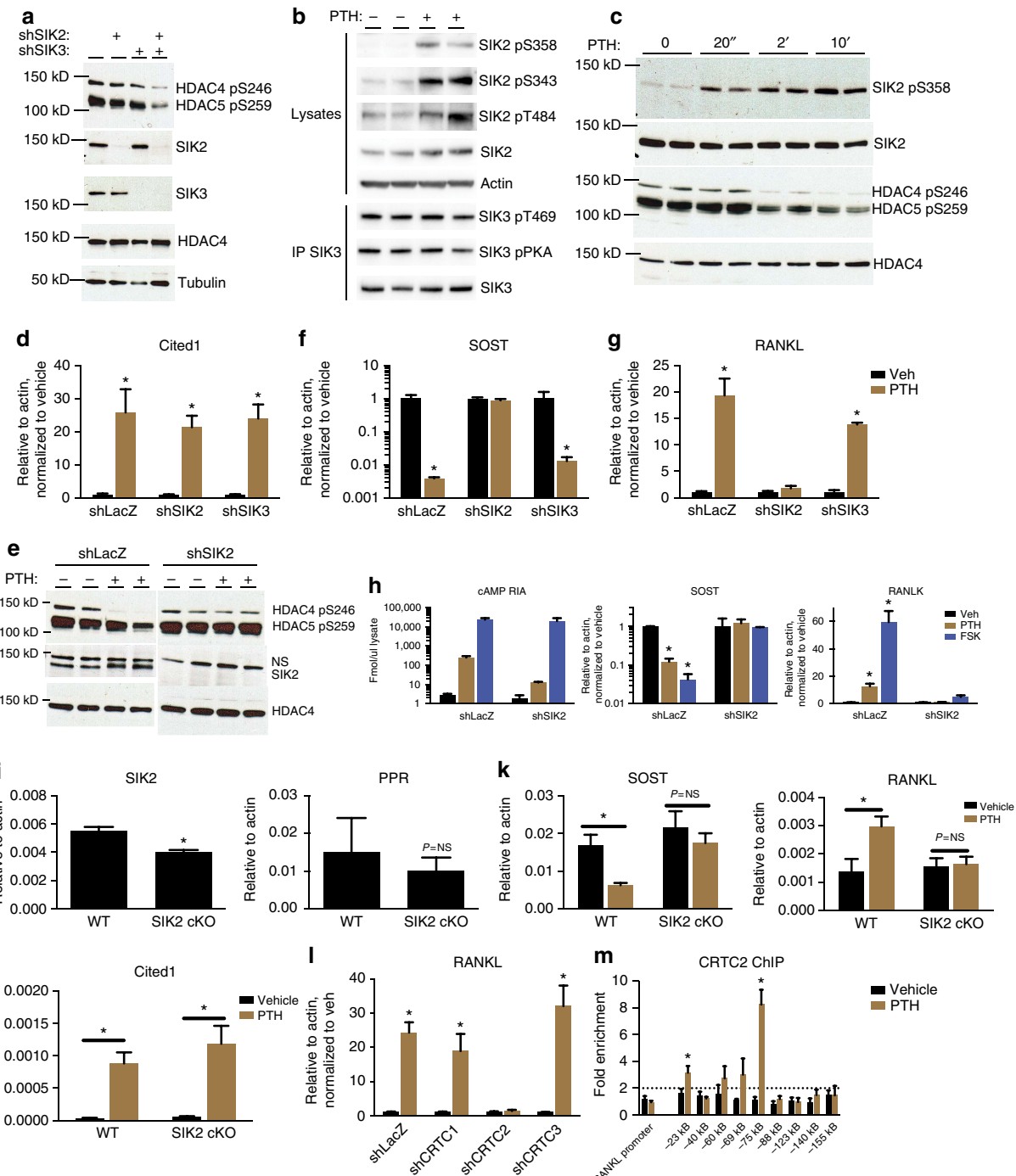

**Figure 4 | SIK2 activity is regulated by PTH signaling.** (**a**) Ocy454 cells were infected with shRNA-expressing lentiviruses, followed by immunoblotting. (**b**) Ocy454 cells were treated with vehicle or PTH (50 nM, 30′), followed by immunoblotting. In the bottom panels, SIK3 immunoprecipitation was performed followed by immunoblotting. (**c**) Ocy454 cells were treated with PTH (50 nM), followed by immunoblotting. (**d**) Ocy454 cells infected with either control, sh*SIK2* or sh*SIK3*-expressing lentiviruses were treated with PTH (1 nM) for 4 h. RNA was isolated and *CITED1* transcript abundance measured. * indicates $P < 0.05$ comparing vehicle and PTH for each cell line. (**e**) Control and sh*SIK2* cells were treated with PTH (50 nM, 30′) and then analysed as in (**b**). (**f,g**) Control, sh*SIK2* and sh*SIK3* cells were treated as in (**d**), and SOST and RANKL transcript abundance measured. * indicates $P < 0.05$ comparing vehicle and PTH. (**h**) Left, control and sh*SIK2* cells were treated with vehicle, PTH (25 nM) or forskolin (FSK, 5 μg ml$^{-1}$) for 30 min followed by cAMP radioimmunoassay. Middle/right, cells were treated with PTH (2.5 nM) or forskolin (500 ng ml$^{-1}$) for 4 h, gene expression was analysed. (**i**) RNA from femurae of 5 week old male WT (*SIK2 f/f*) or *SIK2$^{OcyKO}$* (SIK2 f/f;DMP1-Cre) mice ($n = 3$ group$^{-1}$) was isolated and *SIK2* and PTH receptor (*PPR*) transcripts measured by RT-qPCR. * indicates $P < 0.001$ comparing WT and SIK2 cKO mice. (**j,k**) Mice as in (**i**) were treated with a single dose of PTH (1 mg kg$^{-1}$) and killed 2 h later. Expression of *CITED1*, *SOST* and *RANKL* in femur RNA was determined by RT-qPCR. * indicates $P < 0.05$ comparing vehicle and PTH. (**l**) Ocy454 cells were infected with shRNA-expressing lentiviruses targeting *CRTC1*, *CRTC2*, or *CRTC3*. Cells were then treated with PTH (1 nM, 4 h) and *RANKL* transcript abundance was measured. (**m**) Ocy454 cells were treated with vehicle or PTH (20 nM, 60 min) followed by ChIP for CRTC2. DNA was quantified by qPCR using primers detecting the indicated RANKL regions, and data are expressed as fold enrichment versus control IgG. *indicates $P < 0.05$ comparing vehicle and PTH. The −23 kB and −75 kB enhancers correspond to 'D2' and 'D5' enhancers[42].

(45 genes). The appropriateness of gene categorization was assessed for selected genes from each of these six clusters (*FAM69C*, *ADAMTS1*, *WNT4*, *KLHL30*, *DUSP6* and *CD200*, respectively) by Quantitative reverse transcription PCR (RT-qPCR) from independently-generated samples (Fig. 6c–h).

While YKL-05-093 regulation of many of its target genes not co-regulated by PTH did occur in cells lacking SIK2 and SIK3 (Supplementary Fig. 6E), regulation of *WNT4* and *CD200* (genes co-regulated by both PTH and YKL-05-093) by YKL-05-093 did not occur in SIK2/3 deficient cells (Fig. 6i,j). In total, 13/19 genes

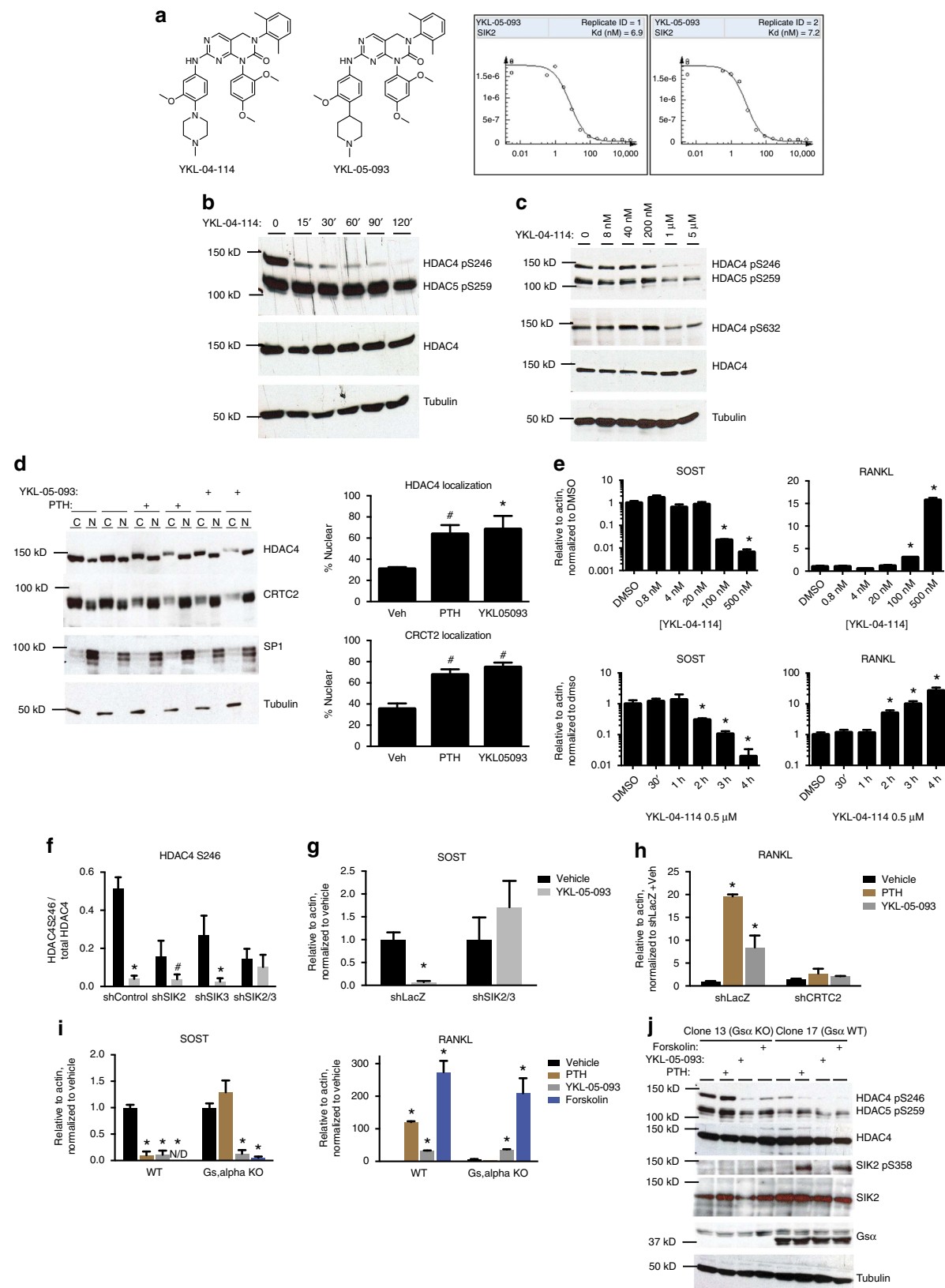

measured showed *SIK2/3*-dependent regulation by YKL-05-093, while 6/19 genes measured showed regulation by YKL-05-093 independent of the presence of *SIK2/3* (Fig. 6i,j, Supplementary Fig. 6E). Taken together, these results demonstrate that a major arm of PTH signalling in Ocy454 cells can be mimicked by SIK inhibition.

**YKL-05-093 mimics PTH actions *in vivo*.** While YKL-04-114 and YKL-05-093 had comparable activity *in vitro,* YKL-05-093 showed improved stability when exposed to murine hepatic microsomes *in vitro* (Supplementary Fig. 7). Therefore, mice were treated with YKL-05-093 and effects on gene expression in bone were assessed 2 h later. Similar to acute PTH administration (Fig. 3), intraperitoneal YKL-05-093 administration led to dose-dependent *SOST* suppression and *RANKL* up-regulation in osteocyte-enriched bone RNA (Fig. 7a,b). This was accompanied by reductions in sclerostin protein levels measured by immunohistochemistry (Fig. 7c). Finally, expression of genes identified by RNA-Seq as co-regulated by PTH and YKL-05-093 *in vitro* were measured: as shown in Fig. 7d–i, *in vivo* 20 umol kg$^{-1}$ YKL-05-093 treatment leads to significant regulation of *VDR*, *WNT4*, *NR4A2*, *NUAK1*, *PDGFA* and *CD200* expression in the directions predicted from the *in vitro* experiments. Therefore, acute YKL-05-093 treatment *in vitro* and *in vivo* engages a program of gene expression quite similar to one used by parathyroid hormone, thus identifying SIK inhibition as an important mechanism used by PTH to regulate gene expression in osteocytes.

**SIK inhibitors boost bone formation and bone mass *in vivo*.** Efforts to determine the skeletal effects of prolonged once daily YKL-05-093 administration were unsuccessful due to toxicity associated with repeated dosing. Therefore, we turned our attention to the recently-described and highly-related compound YKL-05-099 (ref. 23). Developed in parallel efforts to design SIK inhibitors suitable for *in vivo* use, YKL-05-099 is well-tolerated and achieves free serum concentrations above its IC$_{50}$ for SIK2 (34 nM) for >16 h (ref. 23).

First, *in vitro* experiments were performed to characterize the effects of YKL-05-099 in Ocy454 cells. In these experiments, YKL-05-099 was compared side-by-side with YKL-05-093. As expected, YKL-05-099 leads to dose-dependent reduction in HDAC4 S246 phosphorylation (Fig. 8a). Furthermore, YKL-05-099 treatment causes *SOST* down-regulation and *RANKL* up-regulation in a SIK2/3-dependent manner (Fig. 8b). Like YKL-05-093, acute intraperitoneal administration of

YKL-05-099 *in vivo* leads to *SOST* down-regulation and *RANKL* up-regulation (Fig. 8c).

Male mice were then treated with vehicle or YKL-05-099 (6 mg kg$^{-1}$) once daily via intraperitoneal injection for 2 weeks. Bone RNA from these animals revealed that *RANKL* levels were increased and there was a trend towards reduced *SOST* (Fig. 8d). In addition, genes expressed by osteoblasts (osteocalcin (encoded by the *BGLAP* gene) and *COL1A1*) were significantly increased by YKL-05-099 treatment, suggesting possible positive effects on osteoblastic bone anabolism (Fig. 8d). To determine effects on bone mass and cellular composition/activity, static and dynamic histomorphometry were performed. Indeed, once daily YKL-05-099 treatment increased cancellous bone mass (Fig. 8e) and osteoid surface (Fig. 8f), suggesting accelerated bone formation. Dynamic histomorphometry revealed that YKL-05-099 led to increased mineralizing surface, a trend towards increased matrix apposition rate, and increased bone formation rate (Fig. 8g,h,i,l). At the cellular level, YKL-05-099 treatment increased osteoblast numbers (Fig. 8j,m) and reduced osteoclast numbers (Fig. 8k). Other than the observed reduction in osteoclast numbers (see Discussion), these findings are quite similar to the effects of once daily PTH treatment.

### Discussion

PTH is currently the only approved osteoporosis therapy that promotes new bone formation. While its effects on target cells in bone are broad, major target genes in osteocytes responsible for its ability to increase both bone formation and resorption include *SOST* and *RANKL*, respectively. Here we demonstrate that SIKs act as gatekeepers to regulate a major arm of PTH signalling in osteocytes, including (but not limited to) these two important target genes. Tonic SIK activity leads to constitutive phosphorylation and cytoplasmic localization of HDAC4/5 and CRTC2. Activation of protein kinase A, as occurs with activation of the PTH receptor[7], leads to multisite phosphorylation on SIK2, modifications that inhibit its cellular activity[36,38]. This inhibition reduces tonic HDAC4/5 and CRTC2 phosphorylation, which in turn leads to their nuclear localization and action on respective target genes (Fig. 9). Whether this pathway operates in other PTH/PTHrP target cells, such as chondrocytes[45], renal epithelial cells[46], T lymphocytes[47] and adipocytes[48] remains to be determined.

HDAC4/5 are required for PTH-stimulated *SOST* repression in osteocytes, through effects on MEF2C binding to the +45 kB *SOST* enhancer. Previous overexpression studies have suggested that PTH signalling impinges on the upstream *SOST* enhancer[29,49,50]: here we show that HDAC4/5 are required for this effect using loss of function approaches *in vitro* and *in vivo*.

**Figure 5 | SIK inhibitors regulate *SOST* and *RANKL* expression.** (**a**) Structure of YKL-04-114 (left), YKL-05-093 (middle) and YKL-05-093 $K_d$ determination curves for SIK2 (right). For the $K_d$ determination curves, the *y*-axis represents the amount of bound kinase measured by qPCR (see Methods), and the *x*-axis represents the corresponding compound concentration in nM. (**b**) Ocy454 cells were treated with YKL-04-114 (10 μM) for the indicated times, followed by immunoblotting of whole cell lysates as indicated. (**c**) Ocy454 cells were treated with the indicated concentrations of YKL-04-114 for 60 min, followed by immunoblotting of whole cell lysates as indicated. (**d**) Left: Ocy454 cells were treated with vehicle, PTH (50 nM) or YKL-05-093 (10 μM) for 60 min. Cytosol and nuclear fractions were then generated, followed by immunoblotting as indicated. Right: quantification of nuclear fraction (defined as nuclear/total) of HDAC4 or CRTC2. * indicates $P<0.01$ comparing treatment versus vehicle. (**e**) Top: Ocy454 cells were treated with the indicated concentrations of YKL-04-114 for 4 h, followed by RT-qPCR. Bottom: Cells were treated with YKL-04-114 (0.5 μM) for the indicated times. * indicates $P<0.05$ comparing treatment versus vehicle. (**f**) Cells lacking SIK2, SIK3 or both were treated with YKL-05-093 (10 μM for 45 min). Quantification of HDAC4 S246 phosphorylation, as assessed by densitometric analysis of immunoblots, is shown. * indicates $P<0.01$ comparing treatment versus vehicle. # indicates $P<0.05$ for the same comparison. (**g**) Control and *SIK2/3* deficient cells were treated with YKL-05-093 (0.5 μM) for 4 h and *SOST* transcript abundance was measured by RT-qPCR. (**h**) Control and *CRTC2* shRNA cells were treated with PTH (1 nM) or YKL-05-093 (0.5 μM) for 4 h and RANKL transcript abundance was measured by RT-qPCR. (**i**) Control and Gsα-deficient Ocy454 cells were treated with PTH (1 nM), YKL-05-093 (0.5 μM) or forskolin (5 μg ml$^{-1}$) and *SOST* and *RANKL* transcript abundance was determined by RT-qPCR. For (**h**) and (**i**), * indicates $P<0.05$ comparing treatment and vehicle. (**j**) Control and Gsα-deficient Ocy454 cells were treated with PTH (50 nM), YKL-05-093 (10 μM) or forskolin (5 μg ml$^{-1}$) for 30 min. Whole cell lysates were generated and immunoblotted as indicated.

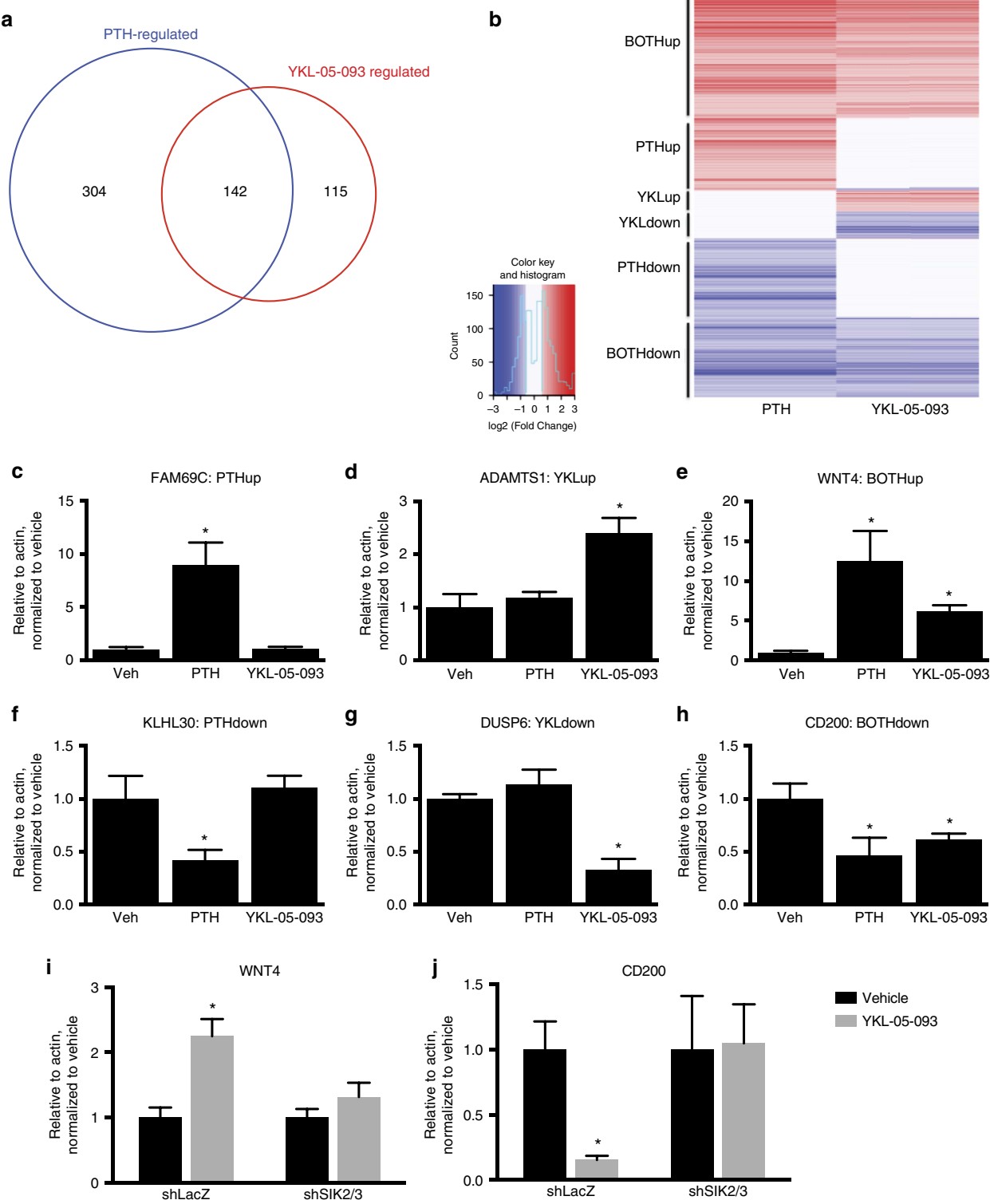

**Figure 6 | YKL-05-093 and PTH similarly affect gene expression.** (**a**) Venn diagram showing overlap between differentially-expressed genes (fold chance > 2, FDR < 0.05) determined by RNA-Seq from Ocy454 cells treated with vehicle, PTH (1 nM) or YKL-05-093 (0.5 μM) for 4 h. (**b**) Heat map showing six different clusters of differentially expressed genes. Each row corresponds to a single differentially expressed gene. Colour coding is with respect to the average log2 (fold change) for each gene comparing treatment to vehicle. Genes were ordered by the strength of the significance of the fold change comparing PTH and vehicle. (**c–h**) Ocy454 cells were treated with vehicle, PTH (1 nM) and YKL-05-093 (0.5 μM) for 4 h, and RT-qPCR was performed for the indicated gene. As described in the text, *FAM69C* and *KLHL30* are regulated by PTH alone, *ADAMTS1* and *DUSP6* are regulated by YKL-05-093 alone, and *WNT4* and *CD200* are regulated by both PTH and YKL-05-093. (**i,j**) Control and *SIK2/3*-deficient cells were treated with vehicle or YKL-05-093 (0.5 μM) for 4 h, and *WNT4* and *CD200* transcript abundance determined by RT-qPCR. For all panels, * indicates *P* < 0.05 comparing vehicle and compound or PTH treatment by student's unpaired two tailed *t*-test.

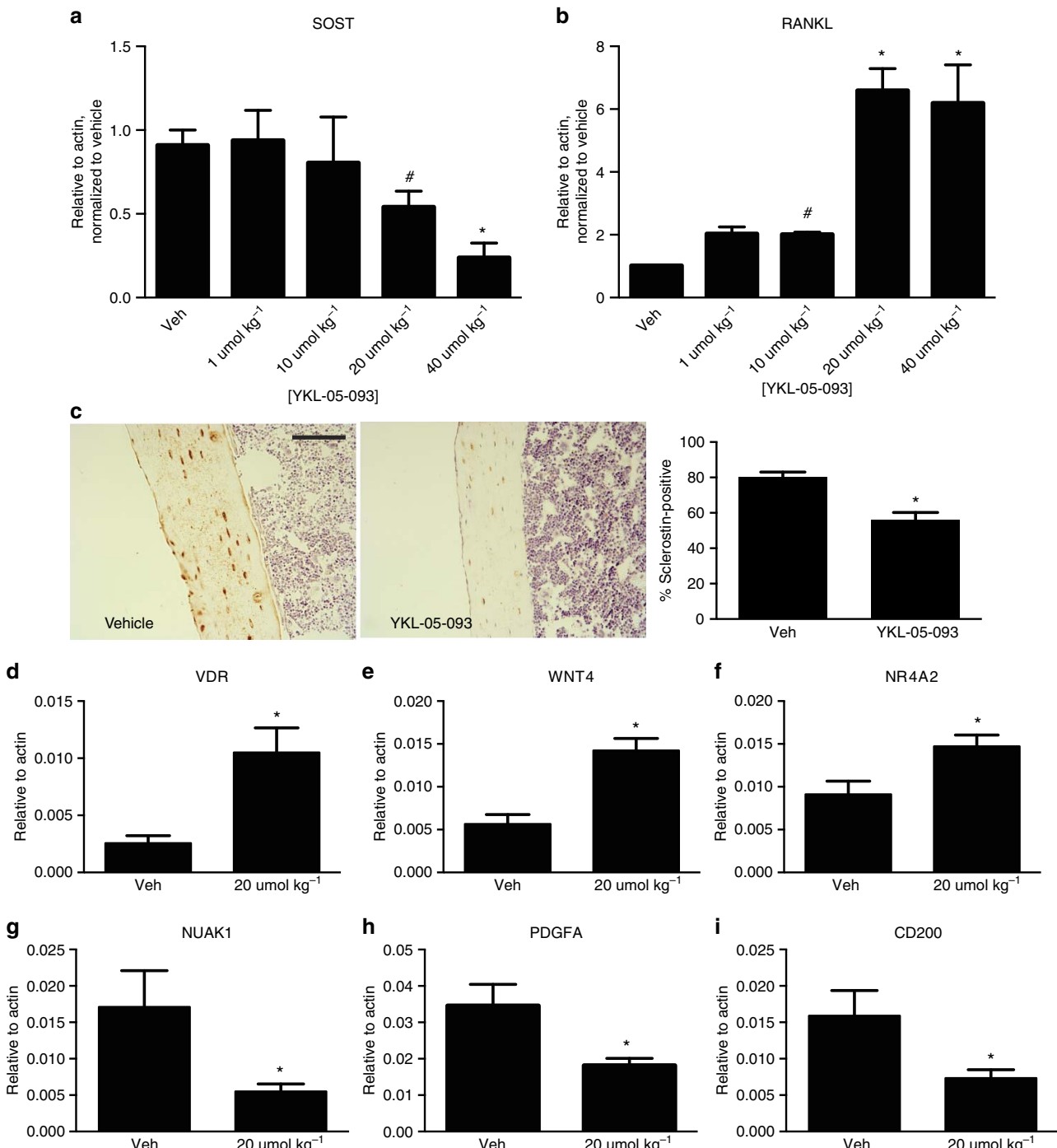

**Figure 7 | Effects of YKL-05-093 administration on bone gene expression *in vivo*. (a,b)** Male mice of 8 week age C57B/6 ($n = 4$ per group) were treated with the indicated dose of YKL-05-093 via intraperitoneal injection. After 2 h, bone RNA was isolated and transcript abundance was measured by RT-qPCR. # indicates $P < 0.05$ versus vehicle and * indicates $P < 0.01$ versus vehicle by student's unpaired two tailed *t*-test. **(c)** Left: sclerostin immunohistochemistry was performed 2 h after intraperitoneal injection with either vehicle or YKL-05-093 (20 µmol kg$^{-1}$). Right: quantification of sclerostin-positive cortical osteocytes, $n = 4$ mice per treatment group, * indicated $P < 0.01$ versus vehicle. Scale bar, 40 µm. **(d-i)** Genes regulated by PTH and YKL-05-093 *in vitro* are also regulated by YKL-05-093 *in vivo*. Male mice were treated with YKL-05-093 (20 µmol kg$^{-1}$) and bone RNA collected 2 h later as in **a**. *indicates $P < 0.05$ versus vehicle.

At later time points, PTH treatment reduces in *MEF2C* mRNA levels[32,33,51], in addition to the post-translational effects on DNA binding observed here earlier (Fig. 2g). Similarly, PTH induces both the rapid nuclear translocation of HDAC4 and, at later time points, increases in *HDAC4* mRNA (Fig. 6 and ref. 52). It is interesting that PTH signalling has evolved two complementary mechanisms to inhibit MEF2C activity: HDAC4/5-mediated

inhibition of binding of MEF2C to target genes and inhibition of transcription of the *MEF2C* gene. Since *MEF2C* autoregulation is known to occur[53], future studies will focus on whether class IIa HDACs regulate MEF2C-driven expression of *MEF2C* itself, and other targets of MEF2C in osteocytes[54].

*HDAC4/5* 'DKO' mice display several phenotypes not present in either single knockout strain or in mice over-expressing

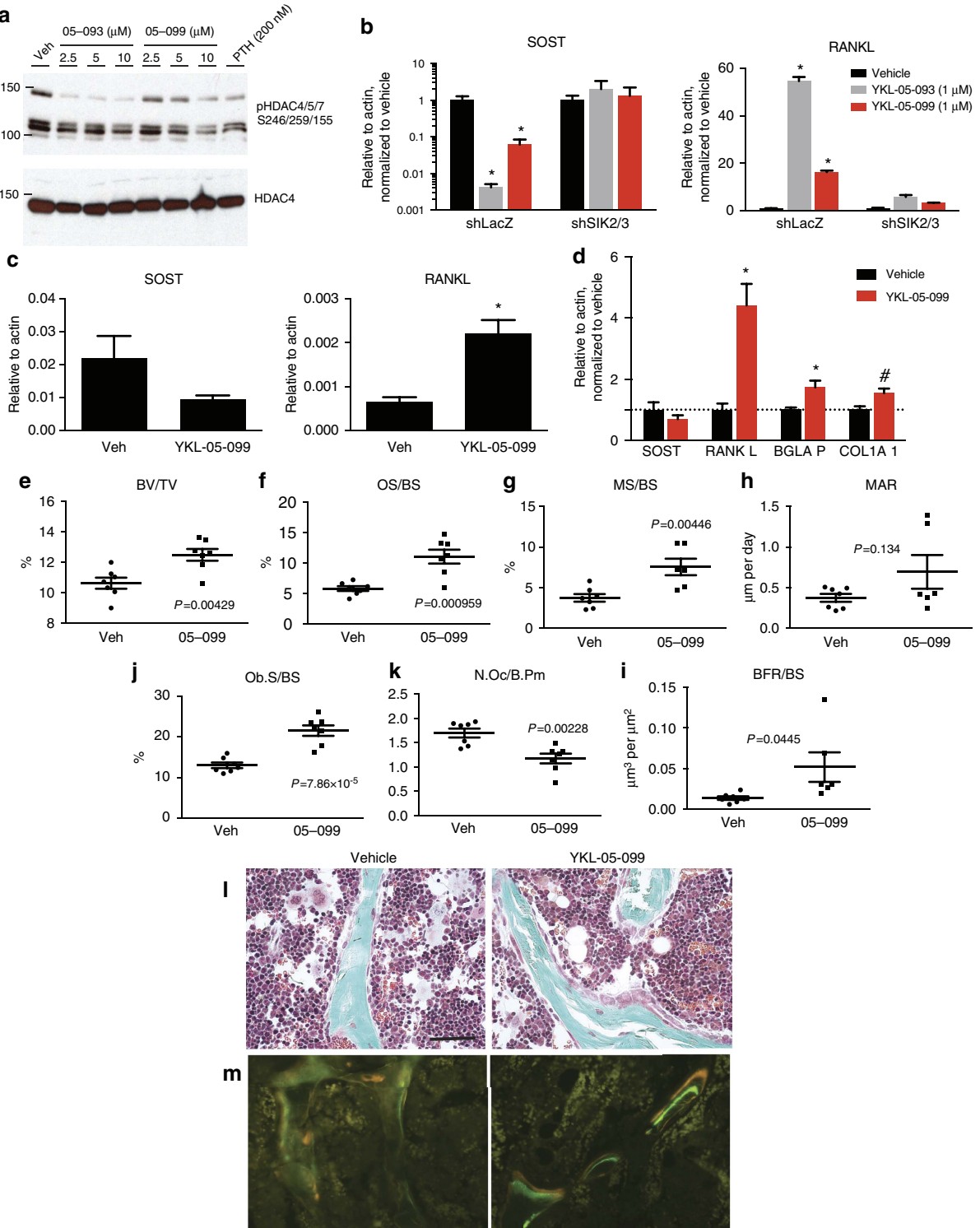

**Figure 8 | YKL-05-099 increases bone formation and bone mass *in vivo*.** (**a**) Ocy454 cells were treated with the indicated doses of YKL-05-093, YKL-05-099 or PTH for 20 min. Whole cell extracts were generated, followed by immunoblotting. (**b**) Control or *shSIK2/3* Ocy454 cells were treated with YKL-05-093 or YKL-05-099 (1 μM) for 4 h. RNA was prepared, and gene expression analysed by RT-qPCR. Both YKL-05-093 and YKL-05-099 regulate *SOST* and *RANKL* expression in control, but not SIK2/3-deficient cells. (**c**) Male mice of 8 week age (*n* = 5 per group) were treated with a single IP dose of YKL-05-099 (20 μmol kg$^{-1}$) or vehicle. After 2 h, animals were killed, RNA was prepared from femurs, and gene expression analysed by RT-qPCR. *SOST* down-regulation was observed in response to YKL-05-099, but the *P*-value for this difference was 0.105. * indicates *P* < 0.01. (**d**) Male mice of 8 week age were treated with vehicle (*n* = 8) or YKL-05-099 (*n* = 7, 10 umol kg$^{-1}$, IP) once daily 5 days per week for 2 weeks. Animals were killed 2 h after the final dose, and RNA from femurs analysed for the indicated genes. *BGLAP* encodes osteocalcin. * indicated *P* < 0.01 versus vehicle, # indicates *P* < 0.05 versus vehicle. (**e–k**), static and dynamic histomorphometry were performed on the tibia from the same mice as in **d**. Each data point represents an individual mouse; *P*-values for each difference are shown on the graph. (**l**) Representative trichrome-stained photomicrograph showing increased osteoblasts on cancellous bone surfaces from YKL-05-099-treated mice. Scale bar, 20 μm. (**m**) Dual calcein/demeclocycline images demonstrating increased mineralizing surface in YKL-05-099-treated mice.

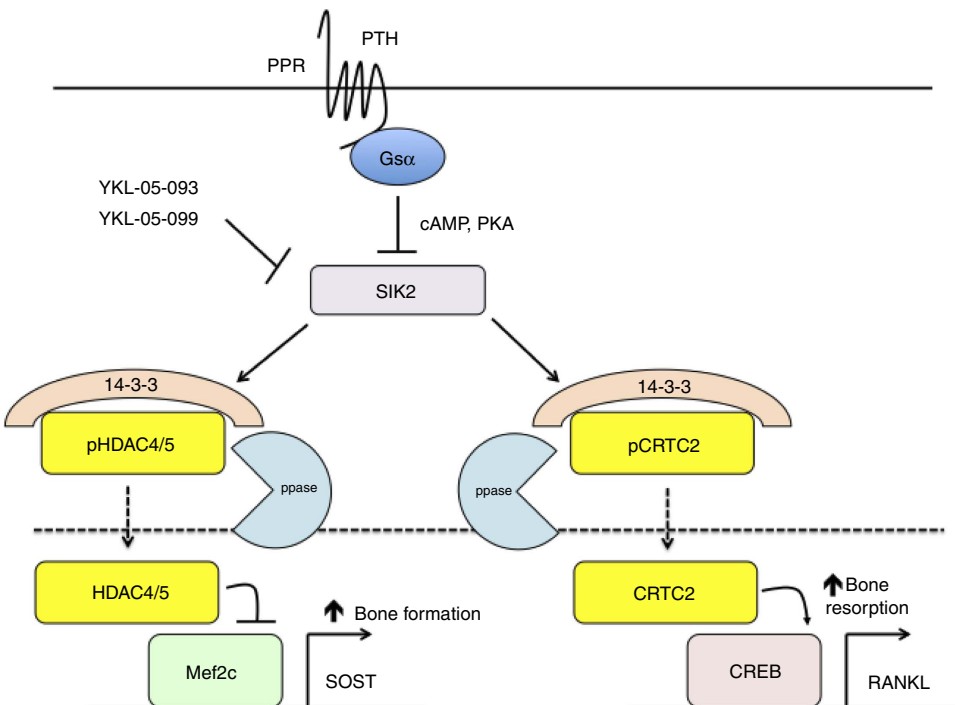

**Figure 9 | Model showing PTH signaling via inhibition of SIK2 in osteocytes.** In the absence of PTH signaling, SIK2 tonically phosphorylates its substrates HDAC4/5 and CRTC2, leading to their cytoplasmic retention via binding to 14-3-3 chaperones. PTH signaling leads to PKA-mediated phosphorylation of SIK2, which inhibits its cellular activity. This in turn reduces phosphorylation of HDAC4/5 and CRTC2, leading to their dephosphorylation by an unknown phosphatase (ppase), and subsequent nuclear translocation. Small molecule SIK inhibitors (YKL-05-093 and YKL-05-099) mimic the effects of PTH by directly blocking SIK2 kinase activity. In the nucleus, HDAC4/5 block MEF2C-driven *SOST* expression, while CRTC2 enhances CREB-mediated *RANKL* gene transcription. PTH-induced reductions in sclerostin contribute to increased bone formation, while PTH-induced increases in RANKL drive increased bone resorption.

sclerostin[55] (Supplementary Fig. 1). Notably, sclerostin transgenic mice do not display woven bone and increased osteocyte density, and sclerostin antibody did not increase BMD in DKO animals. Therefore, class IIa HDACs control expression of additional genes in osteocytes that potently regulate skeletal biology. In addition, as evident by the fact that *HDAC4/5* 'DKO' mice show a preserved bone anabolic effect of intermittent PTH treatment (Supplementary Fig. 4A), class II HDAC/*SOST*-independent pathways that mediate the pharmacologic effects of parathyroid hormone must exist.

Interesting parallels and distinct differences are noted between PTH-mediated *SOST* suppression in osteocytes and PTHrP-mediated suppression of expression of the *Collagen X* gene in growth plate chondrocytes[34]. While both pathways utilize a class IIa HDAC/MEF2 mechanism of action, the signalling events required for HDAC4 nuclear translocation may differ. PTH signalling in osteocytes involves inhibition of SIK activity, while in chondrocytes, PTHrP signalling activates the cAMP-dependent phosphatase PP2A. That being said, a role for SIKs in PTHrP signalling in chondrocytes cannot be excluded given the fact that *SIK3*-deficiency[21] appears to phenocopy the effects of PTHrP overexpression[56]. Our experiments with okadaic acid and *PP2A* shRNA (Supplementary Fig. 4A–D) argue against a major role for PP2A in mediating PTH signalling in osteocytes. Because the inhibition of HDAC4/5 phosphorylation in response to PTH was substantial, any further action of PTH on PP2A or other phosphatases would be likely to have a modest effect on overall phosphorylation levels.

PTH signalling to regulate *RANKL* expression in osteoblasts and osteocytes has been studied extensively over the past decade. Many investigators have demonstrated a role for a cAMP/CREB pathway via the gene's upstream enhancers[13,40,42,57]. Here we

show an additional requirement for the presence of a CREB co-activator, CRTC2, for PTH-induced *RANKL* gene regulation. It is of interest that PTH action requires two pathways, one involving a direct PKA target (CREB) and another that uses PKA-mediated SIK inhibition. Since SIK inhibition, through suppression of *SOST* expression, can also increase bone formation, one can speculate that this use of the SIK pathway 'forces' PTH action to link bone resorption and bone formation.

A recent report has suggested that, in osteoblasts, PTH signalling promotes proteasomal degradation of HDAC4 that in turn allows MEF2C-driven activation of the *RANKL* promoter[24]. We do not observe changes in HDAC4/5 levels after PTH treatment, which may be explained by the differing time courses and cell types used. We favour a model in which PTH induces *RANKL* expression in osteocytes via its −75 kB enhancer through SIK-dependent CRTC2 nuclear translocation.

The use of SIK inhibitors uniquely allows us to examine the acute effects of changes in SIK enzyme activity in cells and mice. These experiments show that the effects of SIK inhibition are rapid enough to mediate the effects of PTH on *SOST* and *RANKL* expression. In this way, though the inhibitors are less specific than gene knockout or shRNA-mediated expression knockdown, their use complements the data derived from the genetic studies. While YKL-05-093 and YKL-05-099 do inhibit kinases other than SIKs when tested *in vitro*, many of their effects in Ocy454 cells, including those on *SOST* and *RANKL* expression, were not observed when SIK2/3 proteins were absent.

The role of *SIK2* and *SIK3* (the predominant SIKs expressed in osteocytes) in bone biology *in vivo* remains incompletely understood. Global *SIK2* knockout mice have been shown to display phenotypes in melanocytes[58], neurons after ischaemic injury[59], cardiomyocytes during hypertrophy[60] and in lipid

homoeostasis[61]. Conditional *SIK2* mutant alleles have been described[38,62] to further study the role of this kinase in hepatocytes and in the pancreas. Global *SIK2* knockout mice have no skeletal phenotype reported to date. Here, we have deleted *SIK2* from DMP1-Cre expressing cells, and have observed that this gene is required for the acute response of osteocytes to PTH. A detailed description of the global bone phenotype of the *SIK2^OcyKO* strain remains to be determined. Global *SIK3* deficient mice display a dramatic growth plate phenotype[21] that confounds study of direct actions of *SIK3* in osteocyte biology *in vivo*. A conditional *SIK3* allele has been reported, and deletion in chondrocytes confirms the cell-intrinsic role for *SIK3* in these cells[63]. No studies to date have examined the role of *SIK3* in osteocytes *in vivo*.

SIK inhibition downstream of cAMP signalling has long been appreciated to occur[36], but the relative contribution of SIK inhibition to overall changes in gene expression due to Gsα-coupled GPCR signalling has not previously been explored. Remarkably, 32% of genes regulated by PTH in osteocytes were co-regulated by YKL-05-093. While it is likely that many of these genes (like *SOST* and *RANKL*) are regulated in turn by HDAC4/5 and CRTC2, undoubtedly additional SIK2/3 substrates may be responsible for these widespread effects.

Recently, pterosin B was reported as a small molecule inhibitor of SIK3 with *in vivo* activity in a SIK3-dependent murine osteoarthritis model[63]. Interestingly, this small molecule leads to ubiquitin-dependent SIK3 degradation, and therefore acts in a manner distinct to that of YKL-05-093 and YKL-05-099, which function as kinase inhibitors[23]. While SIK2 deficiency was sufficient to abrogate responses to parathyroid hormone *in vitro* and *in vivo* (Fig. 4), combined SIK2 and SIK3 deficiency was required to blunt effects of YKL-05-093 and YKL-05-099. This is consistent with potential redundancy between these two kinases[38], and the fact that both inhibitors potently target SIK3 in addition to SIK2.

In many regards, YKL-05-099 treatment mimics the effects of once-daily PTH treatment *in vivo*. However, one notable exception is present. PTH treatment increases osteoclastic bone resorption, in part due to PTH-induced *RANKL* up-regulation[12]. Although YKL-05-099 potently increases *RANKL* levels in bone (Fig. 8d), osteoclast numbers are actually decreased by this treatment (Fig. 8j). In addition to targeting SIK2, YKL-05-099 inhibits the tyrosine kinases Src and CSF-1R (ref. 23). Src deficiency leads to functional osteoclast defects and osteopetrosis[64], and CSF-1R deficiency causes osteoclast-poor osteopetrosis[65]. Therefore, combined SIK and Src/CSF-1R inhibition may lead to the desirable therapeutic combination of increased bone formation and reduced bone resorption. More detailed assessment of the long-term safety profile of YKL-05-099 will be required to determine if its profile of kinase inhibition will be well-tolerated over time.

Recombinant PTH is the only current osteoanabolic therapy approved for osteoporosis treatment. Our data highlight that distinct signalling modules exist downstream of PTH receptor signalling, including a major arm involving SIK inhibition. SIK inhibition is sufficient to reduce sclerostin levels and to mimic many of the other effects of PTH in osteocytes at the level of gene expression. Furthermore, *in vivo* SIK inhibition with YKL-05-099 boosts osteoblast numbers, osteoblast activity and trabecular bone mass. Specific inhibitors of SIK action might provide a novel approach to mimic PTH action to stimulate bone anabolism.

## Methods

**Animal studies.** All animals were housed in the Center for Comparative Medicine at the Massachusetts General Hospital, and all experiments were approved by the hospital's Subcommittee on Research Animal Care. *HDAC5*-null mice[66] and

*HDAC4* f/f mice[67] were generously provided by Dr Eric Olson (University of Texas Southwestern Medical Center, Dallas, TX) and were backcrossed to C57B/6 mice for at least 6 generations. *DMP1-Cre* mice[25] were generously provided by Dr Jian (Jerry) Feng (Texas A&M University, Baylor College of Dentistry, Dallas, TX). 'DKO' HDAC4/5 mice were of the following genotype: *HDAC4f/f;HDAC5^−/−;DMP1-Cre*. *SIK2 f/f* mice were as described[38], and were bred to DMP1-Cre animals to generate *SIK2^OcyKO* mice. ES cells carrying the targeted *SOST* allele Sost^tm1(KOMP)Vlcg, in which the *SOST* coding sequence has been replaced by LacZ and floxed Neo cassette, were obtained from the knockout mouse project (KOMP) repository. Clone VG10069-BE8 was injected into blastocysts, and the resulting *SOST^+/−* mice were crossed to *HDAC5* mutant animals to generate compound heterozygous mice. In all instances, skeletal phenotypes were evaluated in 8 week-old sex-matched littermates. For acute effects of PTH on bone gene expression, animals were treated with PTH (1-34, 300 μg kg^−1, subcutaneous administration) and then killed 90 min later. For acute effects of YKL-05-093 on bone gene expression, animals were treated with the indicated doses of compound (dissolved in PBS + 25 mM HCl) or solvent via intraperitoneal injections and killed 2 h later. Experiments with YKL-05-099 were performed in a similar fashion: compound was dissolved in PBS + 25 mM HCl and injected IP once daily five times per week for a total of 10 injections. For *in vivo* sclerostin antibody treatment, mice were treated twice weekly with sclerostin antibody (50 mg kg^−1, subcutaneous administration, generously provided by Dr Michael Ominsky, Amgen) for 6 weeks. Power calculations were performed based on pilot experiments in which s.d.s and magnitudes of effect sizes were estimated. For experiments in which mice were treated with either vehicle or PTH (or YKL-05-093), mice were assigned to alternating treatment groups in consecutive order.

**Antibodies and compounds.** Antibodies against phospho-HDAC4/5/7 S246/259/155 (3443), phospho-HDAC4 S632 (3424), MEF2C (5030), tubulin (2146), phospho-PKA substrate (9624) and PP2Acs (2259) were purchased from Cell Signaling Technology (Danvers, MA). HDAC4 (ab 12172) and GFP (ab6556) antibodies were from Abcam (Cambridge, MA). FLAG antibody (F1804) was from Sigma (St. Louis, MO). CRTC2 (ST1099) and SP1 (07-645) antibodies were from EMD Millipore (Darmstadt, Germany). Gs,alpha antibody (C-18) was from Santa Cruz Biotechnology (Santa Cruz, CA). Phospho-HDAC5 S279 (ref. 28) antibody was a generous gift from Dr Chris Cowan (McLean Hospital, Belmont, MA). Antibodies recognizing phosphorylated and total forms of SIK2 and SIK3 were as describedin refs 37,38. The phospho-SIK3 (T469) antibody was generated by YenZym Antibodies by immunizing rabbits with mouse SIK3 peptide (Res 463–476 of mouse SIK3 (www.kinase.com): *CLSMRRH-pT-VGVADPR, a terminal cysteine (*C) was added to the peptide sequence to allow peptide conjugation to carrier proteins and 'p'denotes the phosphorylated residue). All antibodies were used at 1:1,000 dilution for immunoblotting. For sclerostin immunohistochemistry, biotinylated anti-sclerostin antibody (BAF1589) was purchased from R + D (Minneapolis, MN). Synthetic human PTH (refs 1–34) was synthesized by Dr Ashok Khatri (peptide/protein core facility, MGH). Forskolin (F6886), staurosporine (S5921) and okadaic acid (O113) were from Sigma. Oligonucleotides were synthesized by the DNA synthesis group of the CCIB DNA Core Facility at MGH (Boston, MA).

**Cell culture.** For all experiments, a single cell subclone of Ocy454 cells[15,16] was used. Cells were passages in alpha-MEM supplemented with 10% heat-inactivated fetal bovine serum and 1% antibiotics (penicillin/streptomycin, Fungizone) at 33 °C with 5% CO_2. Cells were plated at 50,000 cells ml^−1 and allowed to reach confluency at 33 °C (typically in 2–3 days). At this point, cells were transferred to 37 °C for subsequent analysis. For immunoblotting, cells were always analysed after culture at 37 °C for 7 days. For gene expression analysis, cells were analysed after culture at 37 °C for 14 days. Mycoplasma contamination was ruled out by PCR. Cells were routinely assayed for SOST expression at 37 °C and examined for osteocytic morphology.

**shRNA infections and CRISPR/Cas9-mediated gene deletion.** See Supplementary Table 4 for all shRNA and sgRNA targeting sequences used. For shRNA, lentiviruses were produced in 293T cells in a pLKO.1-puro (Addgene, plasmid 8453) backbone. Viral packaging was performed in 293T cells using standard protocols (http://www.broadinstitute.org/rnai/public/resources/protocols). Briefly, 293T cells were plated at 2.2 × 10^5 ml^−1 and transfected the following day with shRNA-expressing plasmid along with psPAX2 (Addgene plasmid 12260) and MD2.G (Addgene plasmid 12259) using Fugene-HD. Medium was changed the next day, and collected 48 h later. For experiments with SIK2/SIK3 double knockdown, one shRNA was transferred into a blasticidin resistance-conferring backbone (Addgene, plasmid 26655). Cells were exposed to lentiviral particles (MOI = 1) overnight at 33 °C in the presence of polybrene (5 μg ml^−1). Media was then changed and puromycin (2 μg ml^−1) and/or blasticidin (4 μg ml^−1) was added. Cells were maintained in selection medium throughout the duration of the experiment. HDAC5 S/A complementary DNA (cDNA) was introduced via lentivirus as described in ref. 16. Briefly, control and Gsα-knockdown cells were infected with lentiviral particles expressing GFP and/or

HDAC5 S249/498A. After 24 h, cells were selected with hygromycin (100 µg ml$^{-1}$) and used for subsequent experiments.

For sgRNA experiments, first Ocy454 cells were stably transduced with a hygromycin resistance-conferring Cas9-expressing lentivirus to ensure no effects on sclerostin secretion. Sclerostin ELISAs were performed exactly as described in ref. 16. For subsequent experiments, sgRNA sequences were subcloned into PX458 (a gift from Dr Feng Zhang, Addgene plasmid 48138 (ref. 68)), a plasmid that co-expressed sgRNA, Cas9 and eGFP. Ocy454 cells were transfected with this plasmid using Fugene HD (Promega, Madison, WI) (1 µg plasmid per well of a six well plate). 48 h later, eGFP$^{hi}$ cells were recovered by FACS-based sorting and plated in 96 well plates at 1 cell per well. Media was changed once weekly, and 3 weeks later colonies were identified by visual inspection. Colonies were then expanded and analysed for loss of target protein expression by immunoblotting. For HDAC4 and Gs,α targeting experiments, at least three independent clones (deriving from two independent sgRNA sequences) were analysed and showed similar results. Allele-specific sequencing of mutant clones was performed by amplifying the genomic region of interest surrounding the targeted site by PCR. PCR products were then TOPO-TA cloned (ThermoFisher), and multiple bacterial colonies sequenced using T7 sequencing primer.

**Real-time quantitative PCR.** Total RNA was extracted from cultured cells using RNeasy (Qiagen, Venlo, Netherlands) following the manufacturer's instructions. For long bone RNA isolation, mice were killed and both femurs were rapidly dissected on ice. Soft tissue was removed and epiphyses cut. Bone marrow cells were then removed by serial flushing with ice-cold PBS. TRIzol (Life Technologies) was added and sampled were frozen at −80 °C and then homogenized. RNA was then extracted per the manufacturer's instructions, and further purified on RNeasy microcolumns before cDNA synthesis. RNA with a A260/280 ratio <1.7 was not used for downstream analysis. For cDNA synthesis, 1 µg RNA was used in synthesis reactions according to the instructions of the manufacturer (Primescript RT, Takara). SYBR Green-based quantitative PCR (qPCR) detection was performed using FastStart Universal SYBR Green (Roche, Basel, Switzerland) on a StepOne Plus (Applied Biosystems, Carlsbad, CA) thermocycler. All PCR primer sequences are listed in Supplementary Table 4.

**Immunoprecipitation and immunoblotting.** Whole cell lysates were prepared using TNT buffer (20 mM Tris–HCl pH 8, 200 mM NaCl, 0.5% Triton X-100 supplemented with 1 mM DTT, 1 mM NaF and protease inhibitors (Pierce, catalogue #88266). This lysis buffer was used for all experiments except those in which SIK2 and SIK3 phosphorylation was measured using phospho-specific antibodies: for those experiments, cells were lysed in buffer containing 50 mM Tris–HCl pH 7.5, 270 mM sucrose, 1 mM EDTA, 1 mM EGTA, 1 mM NaF, 1 mM DTT and protease inhibitors (Sigma, P8340). MEF2C (ref. 16) and SIK3 (refs 37,38) immunoprecipitations were performed as described. Briefly, lysates were precleared with protein A/G, then 0.5 mg total protein incubated with 1 µg antibody overnight at 4 °C. The next morning, immune complexes were precipitated with protein A/G agarose, washed three times in ice-cold lysis buffer, and precipitated proteins eluted by boiling in SDS-sample buffer at 95 °C for 5 min. Subcellular fractionation was performed using a commercially-available kit (Thermo Scientific, product number 78840) following the manufacturer's instructions. Lysates (15–20 µg cellular protein) were separated by SDS–polyacrylamide gel electrophoresis (SDS–PAGE), and proteins were transferred to nitrocellulose. Membranes were blocked with 5% milk in TBST, and incubated with primary antibody overnight at 4 °C. The next day, membranes were washed, incubated with appropriate horseradish peroxidase (HRP)-coupled secondary antibodies, and signals detected with enhanced chemiluminescence (ECL, Pierce). All immunoblots were repeated at least twice with comparable results obtained. Supplementary Figure 8 shows the full blot corresponding to the scanned portions shown in the main text figures.

**Histology and immunohistochemistry.** Formalin-fixed paraffin-embedded decalcified tibia sections from 8 week-old mice were obtained. Sirius red staining was performed using Sirius red and picric acid obtained from Sigma. Sections were visualized under polarized light. Hematoxylin and eosin (H + E) staining was performed on some sections using standard protocols, and osteocyte density was assessed on cortical bone osteocytes in a medium power field 3 mm below the tibial growth plate. For anti-sclerostin immunohistochemistry, antigen retrieval was performed using proteinase K (20 µg ml$^{-1}$) for 15 min. Endogenous peroxidases were quenched, and slides were blocked in TNB buffer (Perkin-Elmer), then stained with anti-sclerostin antibody at a concentration of 1:200 for 1 h at room temperature. Sections were washed, incubated with HRP-coupled secondary antibodies, signals amplified using tyramide signal amplification and HRP detection was performed using 3,3′-diaminobenzidine (DAB, Vector) for 2–3 min. Slides were briefly counterstained with hematoxylin before mounting. Quantification of sclerostin positive osteocytes was performed on a blinded basis. All photomicrographs were taken 3 mm below the growth plate on the lateral side of the tibia. All osteocytes were counted and then scored as either sclerostin-positive or negative. Sections from at least four mice per experimental group were analysed. Quantification of immunostaining was done based on coded sample numbers in a completely blinded manner. Representative photomicrographs are displayed next to quantification in data figures.

**Chromatin immunoprecipitations.** ChIP assay was performed using a kit (EZ-Chip, Miilipore, 17-371, Billerica, MA) according to the manufacturer's instructions. Briefly, cells were grown at 37 °C for 7 days, followed by PTH treatment (25–50 nM) for the indicated times. Cells were then cross-linked with 1% formaldehyde for 10 min and then quenched with 0.125M glycine. Cells were lysed and sonicated with 10 pulses for 30 s each to fragment DNA to 200–800 bp fragments. DNA-protein complexes were precipitated using 1.5 µg antibodies (MEF2C, CRTC2 or control rabbit IgG) overnight at 4 °C. Immune complexes were precipitated, DNA was purified and real-time PCR was conducted using primer sets (Supplementary Table 4) to detect the +45 kB SOST enhancer and upstream RANKL enhancers. Data are expressed as relative enrichment for each antibody (above control IgG) for each primer set. Data shown represent triplicate biological repeats within experiments, and each experiment was performed at least twice.

**cAMP radioimmunoassay.** Cells, in 96 well plates, were treated with indicated ligands for 20 min at room temperature in the presence of the phosphodiesterase inhibitor 3-isobutyl-1-methylxanthine (IBMX, Sigma I5879, 2 mM). The medium was then removed and cells were lysed in 50 mM HCl and transferred to −80 °C. Thawed lysates were diluted 1:5 with dH$_2$O, and an 10 µl aliquot was assessed for cAMP content by radioimmunoassay using $^{125}$I-cAMP analogue as a tracer and unlabelled cAMP to generate a standard curve.

**RNA-sequencing.** Total RNA was subjected to ribosomal RNA (rRNA) depletion using RiboZero kit (Illumina) followed by NGS library construction using NEBNext Ultra Directional RNA Library Prep Kit for Illumina (New England Biolabs). Experimental duplicates were performed for each condition. Sequencing was performed on Illumina HiSeq 2500 instrument, resulting in an average of 33 million pairs of 50 bp reads per sample. Sequencing reads were mapped to the mouse reference genome (mm10/GRCm38) using STAR (http://bioinformatics.oxfordjournals.org/content/early/2012/10/25/bioinformatics.bts635). Gene expression counts were calculated using HTSeq v.0.6.0 (http://www.huber.embl.de/users/anders/HTSeq/doc/overview.html) based on a current Ensembl annotation file for mm10/GRCm38 (release 75). Differential expression analysis was performed using EgdeR package based on the criteria of more than two-fold change in expression value versus control and false discovery rates (FDR) <0.05. Venn diagrams from gene set analysis were generated using genes with >1.5 fold change in expression values and FDR <0.05. Significance testing for gene set overlap was performed according to a standard hypergeometric distribution, $P$-values $<2.2 \times 10^{-16}$. The RNA-seq data are deposited in GEO under accession code GSE76932.

**Small molecule synthesis.** Details regarding synthesis of YKL-04-114 and YKL-05-093 are found in Supplementary Methods. Supplementary Fig. 9 shows nuclear magnetic resonance (NMR) spectra for these compounds and related intermediates.

***In vitro* kinase assays.** ScanEDGE kinase assays panelling specificity across a panel of 96 representative kinases were performed by DiscoverX (Fremont, CA). For most assays, kinase-tagged T7 phage strains were grown in parallel in 24-well blocks in an *E. coli* host derived from the BL21 strain. Bacteria were grown to log phase and infected with T7 phage from frozen stock (MOI = 0.4) and incubated with shaking at 32 °C until lysis (90–150 min). The lysates were centrifuged (6,000*g*) and filtered (0.2 µm) to remove cell debris. The remaining kinases were produced in HEK-293 cells and subsequently tagged with DNA for qPCR detection. Streptavidin-coated magnetic beads were treated with biotinylated small molecule ligands for 30 min at room temperature to generate affinity resins for kinase assays. The liganded beads were blocked with excess biotin and washing with blocking buffer (SeaBlock (Pierce), 1% BSA, 0.05% Tween 20, 1 mM DTT) to remove unbound ligand and to reduce non-specific phage binding. Binding reactions were assembled by combining kinases, liganded affinity beads, and test compounds in 1 × binding buffer (20% SeaBlock, 0.17 × PBS, 0.05% Tween 20, 6 mM DTT). Test compounds were prepared as 40 × stocks in 100% DMSO and directly diluted into the assay. All reactions were performed in polypropylene 384 well plates in a final volume of 40 µl. The assay plates were incubated at room temperature with shaking for 1 h and the affinity beads were washed with wash buffer (1 × PBS, 0.05% Tween 20). The beads were then resuspended in elution buffer (1 × PBS, 0.05% Tween 20, 0.5 µM of the non-biotinylated affinity ligand) and incubated at room temperature with shaking for 30 min. The kinase concentration in the eluates was measured by qPCR. YKL-05-093 was screened in this assay at 71 nM (ten times its $K_d$ for SIK2), and results are reported as '% control', where lower numbers indicate stronger hits.

**Micro-CT.** Assessment of bone morphology and microarchitecture was performed with high-resolution micro–computed tomography ($\mu$CT40; Scanco Medical, Brüttisellen, Switzerland). In brief, the distal femoral metaphysis and mid-diaphysis were scanned using 70 kVp peak X-ray tube potential, 113 mAs X-ray tube current, 200 ms integration time, and 10-$\mu$m isotropic voxel size. Cancellous bone was assessed in the distal metaphysis and cortical bone was assessed in the mid-diaphysis. The femoral metaphysis region began 1,700 $\mu$m proximal to the distal growth plate and extended 1,500 $\mu$m distally. Cancellous bone was separated from cortical bone with a semiautomated contouring program. For the cancellous bone region we assessed bone volume fraction (BV/TV, %), trabecular thickness (Tb.Th, mm), trabecular separation (Tb.Sp, mm), trabecular number (Tb.N, 1 mm$^{-1}$), connectivity density (Conn.D, 1 mm$^{-3}$), and structure model index. Transverse CT slices were also acquired in a 500 $\mu$m long region at the femoral mid-diaphysis to assess total cross-sectional area, cortical bone area, and medullary area (Tt.Ar, Ct.Ar and Ma.Ar, respectively, all mm$^2$); bone area fraction (Ct.Ar/Tt.Ar, %), cortical thickness (Ct.Th, mm), porosity (Ct.Po, %) and minimum ($I_{\min}$, mm$^4$), maximum ($I_{\max}$, mm$^4$) and polar ($J$, mm$^4$) moments of inertia. Bone was segmented from soft tissue using the same threshold, 300 mg HA cm$^{-3}$ for trabecular and 733 mg HA cm$^{-3}$ for cortical bone. Scanning and analyses adhered to the guidelines for the use of micro-CT for the assessment of bone architecture in rodents[69]. For the primary spongiosa region (where intermittent PTH treatment has its predominant effect) analysed in Supplementary Fig. 4A, coronal CT slices were evaluated in a 500 $\mu$m (50 slices) region located centrally in the bone. The region of interest began 1000 $\mu$m superior to the epiphysis and included all primary spongiosa and the medullary cavity. The primary spongiosa bone region was identified by semi-manually contouring the region of interest. Images were thresholded using an adaptive-iterative algorithm. The average adaptive-iterative threshold of control mice (WT, vehicle treated) for the region of interest (299 mgHA cm$^{-3}$) was then used to segment bone from soft tissue for all distal femur images. Micro-CT analysis was done in a completely blinded manner with all mice assigned to coded sample numbers.

**Histomorphometry.** Right tibia from 8-week-old mice were subjected to bone histomorphometric analysis. The mice were injected with 20 mg kg$^{-1}$ body weight of calcein and 40 mg kg$^{-1}$ body weight of demeclocycline on 7 and 2 days before necropsy, respectively. The tibia was dissected and fixed in 70% ethanol for 3 days. Fixed bones were dehydrated in graded ethanol, then infiltrated and embedded in methylmethacrylate without demineralization. Undecalcified 5 $\mu$m and 10 $\mu$m thick longitudinal sections were obtained using a microtome (RM2255, Leica Biosystems., IL, USA). The 5 $\mu$m sections were stained with Goldner Trichome and at least two nonsecutive sections per sample were examined for measurement of cellular parameters. The 10 $\mu$m sections were left unstained for measurement of dynamic parameters, and only double-labels were measured, avoiding nonspecific fluorochrome labelling. A standard dynamic bone histomorphometric analysis of the tibial metaphysis was done using the Osteomeasure analysing system (Osteometrics Inc., Decatur, GA, USA). Measurements were performed in the area of secondary spongiosa, 200 $\mu$m below the proximal growth plate. The observer was blinded to the experimental genotype at the time of measurement. The structural, dynamic and cellular parameters were calculated and expressed according to the standardized nomenclature[70].

**Statistics.** All experiments were performed at least twice. Data are expressed as means of triplicate biological repeats within a representative experiment plus/minus standard error. Statistical analyses was peformed using an unpaired two-tailed Student's $t$-test (Microsoft Excel), $P$-values $< 0.05$ were considered to be significant. Variation between groups was similar in all cases.

**Data availability.** The RNA-seq data are deposited in GEO under accession code GSE76932. The authors declare that all other data supporting the findings of this study are available within the article and its supplementary information files.

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

## Acknowledgements

We thank Drs David Fisher, Michael Ominsky, Kelly Lauter, Chris Cowan, Eric Olson, Jerry Feng, Alykhan Shamji and John Doench for providing reagents and suggestions, Dr Sean Wu for performing blastocyst injections, and Drs Michael Mannstadt, Joseph Avruch and Tatsuya Kobayashi for comments on this manuscript. Sclerostin antibody was generously provided by Amgen Inc. and UCB Pharma. We thank for MGH Endocrine Unit Center for Skeletal Research P30 core (P30AR066261) for assistance. This work is supported by grants from the Ellison Foundation (HMK) and the NIH: K08AR067285 (MNW), R03AR059942 (JYW), P01DK011794 (HMK) and P30AR066261 (HMK).

## Author contributions

M.W. and H.K. designed all experiments and wrote the manuscript. Y.L., J.W. and N.G. synthesized small molecules, and assisted in design and interpretation of all experiments involving their use. Experiments were performed by M.W., O.G., E.W., N.G., B.B., M.O.M. and B.C. D.B. and M.B. performed and analysed micro-CT data. K.N., J.M. and R.B. performed and analysed histomorphometry data. A.A. and R.S. analysed RNA-seq data. T.G. assisted with cAMP radioimmunoassay interpretation. S.N., J.W., K.S., M.F., T.S., R.X. and P.D.P. provided reagents and planned experiments. All authors edited and approved the manuscript.
