## [Peer Review File · Nature Communications]

Reviewers' comments:

Reviewer #1

Expert in bone, HDACs

(Remarks to the Author):

This new manuscript by Wein and colleagues provides evidence that SIK2/3 control osteocyte responses to PTH. The results are interesting because they provide a better mechanistic understanding of how PTH controls sclerostin expression by regulating the activity of Hdac4, Hdac5 and Mef2. They experiments also demonstrate how PTH regulates RANKL expression through CRTC2. The work is of high importance to the field because PTH, RANKL antagonists, and Sclerostin antagonists are therapeutic strategies to treat low bone mass and prevent osteoporosis-related fractures. The experimental plan is rigorous and includes various inhibition strategies (shRNA, CRISPR, CKO and KO mouse models, new pharmacological inhibitors) and a variety of molecular outcomes to test hypotheses. The work is compelling, but a few more pieces of data would complete the story from a mechanistic standpoint.

1. Some studies suggest that Hdac4 and Hdac5 are phosphorylated in the nucleus. This promotes their association with 14-3-3 proteins and export to the cytoplasm. Several nuclear/cytoplasmic fractionation studies were performed in this manuscript. It would be useful to show where SIK2 and SIK3 are localized in these experiments. Some immunofluorescence studies examining co-localization of SIKs with Hdacs might also reveal insights into mechanisms at play.

2. It is interesting that Hdac4 is one of the genes that is upregulated by both PTH and YKL-05-093 and that Hdac9 is repressed by both. In some of the westerns, Hdac4 might be unregulated, whereas in other westerns it is difficult to tell, possibly because of slight overexposure. The authors should comment on this.

3. The N/C fractionation in Figure 5D is not as clean as in the other experiments. Is there a better experiment that could be shown? The authors should also find a way to normalize levels of Hdac4 and CRTC2 to a control protein.

4. The synthesis and early testing of YKL-05-093 is impressive. Supplementary Table 2 indicates that other kinases may also be affected. Because specificity is a potential issue, some in vivo and in vitro studies would be helpful. Does YKL-05-093 affect bone density or other bone properties in mice and bone formation in cell culture models?

5. Figure 6 only shows data for a select group of genes. All the data should be shown, even negative data, in the supplemental figures.

6. The results in Figures 7A and 7B are very stunning due to the rapid disappearance of SOST. Do RANKL levels also change? In Figure 3C/D, were the IHC assays started within 90 minutes of PTH administration?

7. On page 7, the sentence mentioning the "+45kB downstream enhancer (figure 2G)" is confusing. Does that refer to the Sost or Mef2C enhancer?

8. Information on how phosphorylation affects Hdacs should be moved from the results section (page 6) to the introduction (page 4) for uninformed readers. On this same theme, the model in Figure 8 is incorrect if Hdac4/5 and CRTC2 are phosphorylated in the nucleus and then exported to the cytoplasm.

Reviewer #2

Expert in SIKs

(Remarks to the Author):

In the submitted manuscript, the authors demonstrate the PTH-GS α -cAMP-HDAC 4/5-MEF2C-SOST axis in Ocy454 cells in vitro and then go on to show that HDAC 4/5 is responsible for the PTH-induced down-regulation of SOST expression by administering PTH to HDAC 4/5 DKO mice in vivo (Fig. 3). The down-regulation was inhibited by the deletion of both HDAC4 and HDAC5 in mice. Based on this result and the previously reported findings that class II HDACs are substrates of SIKs and that cAMP inhibits SIK2 through PKA-mediated phosphorylation in adipocytes and hepatocytes, the authors hypothesized that SIKs mediate PTH function in osteocytes. Supporting this hypothesis, the authors demonstrate that SIK2 mediates the PTH-induced down-regulation of SOST expression in Ocy454 cells. In addition, the authors found that SIK2 also mediates the PTH-induced up-regulation of RANKL expression in Ocy454 cells. It has been known that CREB activates RANKL transcription and that SIKs inhibit CREB function by phosphorylating CRTC, a co-activator of CREB. The authors demonstrate that PTH promotes the translocation of CRTC2 to the nucleus and that CRTC2 is responsible for the PTH-induced up-regulation of RANKL expression. Finally, the authors investigate the role of SIKs on the PTH regulation of SOST and RANKL mRNA expressions in mice. For this purpose, they identified two small molecules, YKL-04-114 and YKL-05-093, that inhibit SIK by dephosphorylating pS358 (Fig. 5J). YKL regulated the SOST and RANKL mRNA expressions in a SIKs- and CRTC2-dependent manner, respectively, in Ocy454 cells. RNA seq analysis showed that YKL mimics PTH. Finally, treatment of mice with YKL decreased SOST expression and increased RANKL expression in osteocytes.

This work has the potential to significantly advance our understanding of PTH action on osteocytes, which has important implications biologically and medically. Of special note is the elegant work that shows SIKs are substantial mediators of PTH. Another strong point of this study is the identification of YKLs as inhibitors of SIKs and mimics of PTH action in osteocytes. While the findings of the paper are significant advance, several points need to be clarified.

Major

1. The pathway analyses were initially done in Ocy454 cells in vitro. The authors then confirmed the function of HDAC 4/5 in the PTH-induced down-regulation of SOST in mice with HDAC 4/5 DKO background in vivo. Further experiments showed that SIKs are mediators of PTH in Ocy454 cells in vitro and in mice treated with YKL in vivo. As shown by the authors, YKL also affected other enzymes in addition to SIKs. Although the authors show YKL action in Ocy454 cells via SIKs in vitro, it is not known whether this mechanism is also the case in vivo. Experiments such as treating SIK KO or cKO mice with PTH would confirm the authors' conclusion.

2. This reviewer would expect SIK inhibitor compounds, such as YKL, could cure osteoporosis. Such a finding would increase the significance of this study. What is the bone phenotype of mice treated with YKL for certain period of time? Does it mimic the phenotype of mice treated with PTH? It is known that animals treated with PTH show specific histologies in the cortex and spongiosa of bone.

Specific

1. Fig. 2D. Should HDAC4 pS259 be HDAC5 pS259? Why did the amount of HDAC5 pS259 increase with the addition of PTH or Forskolin in clone #11?

2. Sup. Fig. 2E. The amount of HDAC5 pS259 did not decrease with the addition of PTH or FSK.

3. Both shSIK2 and shSIK3 were necessary for substantial dephosphorylation of HDAC 4/5 (Fig. 4A), but PTH did not induce phosphorylation of SIK3 (Fig. 4B). In addition, shSIK2 suppressed both the PTH-induced dephosphorylation of HDAC 4/5 (Fig. 4E) and the PTH-induced down-regulation of SOST expression (Fig. 4F). Does SIK3 also play a role in the changed expression levels of pHDAC 4/5, SOST, and RANKL that are regulated by PTH?

4. Did the authors try to examine the effects of inactivated SIKs or the addition of YK-04-114 on the phosphorylation levels of CRTCs? Are they like the effects seen on HDACs?

5. shSIK2 was sufficient to suppress the PTH effects on SOST (Fig. 4F) and RANKL (Fig. 4G). Both shSIK2 and shSIK3, however, were needed to suppress the YKL-induced down-regulation of SOST

transcription (Figs. 5F, G). Please explain the discrepancy between these two results. In other words, what role does SIK3 play? In addition, how about regulation of RANKL?

6. Fig. 5A graph. Please explain the X- and Y-axes.
7. How does YKL inhibit SIKs? The authors show it dephosphorylates SIK2 pS358 (Fig. 5J). Does YKL act on SIKs directly or does it act on upstream molecules that ultimately phosphorylate SIK2 S358?
8. Please describe the route by which mice were treated with sclerostin antibody, PTH and YKL. Subcutaneous injection or oral administration?
9. Fig. 7C. For the immunohistochemical analysis, Please indicate how many hours after treatment with YKL the sclerostin was examined.
10. What are the bone phenotypes of mice lacking SIKs? Have they been reported?
11. Discussion, first paragraph, last sentence. "Whether this pathway operates in other extraskelatal PTH/PTHrP target cells, such as renal epithelial cells (45), T lymphocytes (46), and adipocytes (47) remains to be determined." Chondrocytes should be included in the list of cells.
12. Discussion, page 16, second paragraph. "While both pathways utilize a class IIa HDAC/MEF2 mechanism of action..." A class IIa HDAC/MEF2 mechanism should be viewed as a SIKs/a class IIa HDAC/MEF2 mechanism and discussed in this respect, because chondrocytes have been reported to use SIK3 (Sasagawa et al., Development 139, 1153-1163, 2012).

Reviewer #3

Expert in bone remodelling, Wnt signalling
(Remarks to the Author):

Wein et al studied the mechanism through which PTH suppresses Sost and stimulates rankl transcription in osteocytes. They have provided evidence that PTH inhibits SIK2 activity to promote nuclear entry of Hdac4/5 in the case of Sost suppression and that of CRTC2 in the case of Rankl induction. Furthermore, the authors developed novel SIK inhibitors to demonstrate the role of SIK in the regulation of Sost and Rankl. Overall, the biochemical evidence for such regulation is strong. However, the overall contribution of such a mechanism to the bone anabolic effect of PTH needs to be clarified to strengthen the paper. Detailed comments are as follows.

- 1) The authors open the paper with supplemental figure1 to argue for the functional interaction of Hdac4/5 and Sost. However, because hdac5^{+/-} does not have a bone phenotype, Fig. S1A does not support that hdac5 works through Sost but rather that the Sost^{+/-} phenotype requires intact hdac5. In Fig. S1D, the authors should use anova to assess statistical significance of the interaction between anti-Sost and Hdac5 deletion. Finally, do the authors have evidence that anti-Sost rescues the bone phenotype in Hdac4/5 double mutant?
- 2) Have the authors tested potential anabolic effect of PTH in Hdac4/5 mutants?
- 3) Does the SIK inhibitor have a bone anabolic effect?
- 4) Fig. S4E showed great decrease in cAMP induction by PTH in SIK2 knockdown cells. This result weakens the argument that the loss of Sost and Rankl response to PTH was strictly due to SIK2 deficiency. Did the authors test the effect of forskolin on Sost/Rankl expression in these cells?

Reviewer #4

Expert in organic chemistry synthesis
(Remarks to the Author):

This reviewer has been asked to focus on the chemical synthesis aspects of the paper, so all comments are limited to this area of the work only.

As part of this manuscript the authors report the synthesis of YKL-05-093 and YKL-04-114. The procedures are sufficiently detailed to allow for a peer to replicate the work described. However the analytical information reported for each compound is insufficient to confirm both the identity

and/or purity of the compounds (in line with normal organic synthesis practice). This must be rectified before this article can be considered for publication.

For compound 2: This is a known compound and reaction (Brossmer,R.; Roehm,E. Justus Liebigs Annalen der Chemie, 1966 , vol. 692, p. 119 - 133, along with numerous modern patent references). References must be provided for this compound.

As a known compound the minimum acceptable analytical data to be reported is: ¹H NMR, ¹³C NMR, IR, Low res MS (given in this case) and melting point (referenced against that known, in this case 52-53 C)

For compounds 3,4,5,YKL-05-093 and YKL-04-114 as unknown compounds the minimum acceptable analytical data to be reported is: ¹H NMR, ¹³C NMR, IR, Low res MS, melting point, High res MS OR elemental analysis (to confirm formula). Purity needs to be demonstrated, this is normally done through inclusion of the ¹H and ¹³C NMR spectra in the supplemental information (additional data such as LC maybe included).

Additional points;

1) For compound 5 where ¹H NMR is given. The authors report the benzylic methylene as 4.74 (dd, J = 5.5, 1.6 Hz, 2H), which looks odd. Can the authors confirm this is not either a mixture of rotamers or a miss-assigned geminally coupled system.

2) For compounds YKL-05-093 and YKL-04-114 the geminal coupling for the methylene in the dihydropyrimidinone ring suggests that these protons are in very different environments. Considering the structures of YKL-05-093 and YKL-04-114 have the authors considered that there may be restricted rotation around the C-N bond between the dihydropyrimidinone ring and the right hand dimethylresorcinol? This molecule may exist as a racemic mixture of atropisomers (this could be examined by chiral HPLC or possibly x-ray crystallography). This should be tested as it is possible that one enantiomer could have improved biological activity.

Reviewer #1

Expert in bone, HDACs

(Remarks to the Author):

This new manuscript by Wein and colleagues provides evidence that SIK2/3 control osteocyte responses to PTH. The results are interesting because they provide a better mechanistic understanding of how PTH controls sclerostin expression by regulating the activity of Hdac4, Hdac5 and Mef2. The experiments also demonstrate how PTH regulates RANKL expression through CRTC2. The work is of high importance to the field because PTH, RANKL antagonists, and Sclerostin antagonists are therapeutic strategies to treat low bone mass and prevent osteoporosis-related fractures. The experimental plan is rigorous and includes various inhibition strategies (shRNA, CRISPR, CKO and KO mouse models, new pharmacological inhibitors) and a variety of molecular outcomes to test hypotheses. The work is compelling, but a few more pieces of

data would complete the story from a mechanistic standpoint.

1. Some studies suggest that Hdac4 and Hdac5 are phosphorylated in the nucleus. This promotes their association with 14-3-3 proteins and export to the cytoplasm. Several nuclear/cytoplasmic fractionation studies were performed in this manuscript. It would be useful to show where SIK2 and SIK3 are localized in these experiments. Some immunofluorescence studies examining co-localization of SIKs with Hdacs might also reveal insights into mechanisms at play.

This is a very important point. Although our SIK2 and SIK3 antibodies do not reliably detect these endogenous proteins for immunocytochemical staining, they do work quite well to detect endogenous SIK2 and SIK3 in osteocytes by immunoblotting. Therefore, in new Supplemental Figure 4E of the revised manuscript, we show subcellular fractionation results indicating that the majority of SIK2 and SIK3 in Ocy454 cells are present in the cytoplasm. This localization pattern resembles that of HDAC4 and HDAC5, and supports our model in which SIK-mediated N-terminal HDAC4/5 phosphorylation occurs in this subcellular compartment.

2. It is interesting that Hdac4 is one of the genes that is upregulated by both PTH and YKL-05-093 and that Hdac9 is repressed by both. In some of the westerns, Hdac4 might be unregulated, whereas in other westerns it is difficult to tell, possibly because of slight overexposure. The authors should comment on this.

HDAC4 mRNA up-regulation by PTH has been described by others (1), consistent with our RNA-seq findings reported here. Across multiple experiments we do observe mild HDAC4 up-regulation at the protein level in response to PTH and YKL-05-093. However, the kinetics of this up-regulation (hours) lag far behind the kinetics of PTH- and YKL-05-093-induced HDAC4 S246 dephosphorylation (seconds/minutes, see Figure 4C). Therefore, while PTH-induced HDAC4 mRNA and protein up-regulation do occur, we favor a model in which the acute effects of PTH and SIK inhibitors are via a subcellular redistribution of HDAC4. A statement to this regard has been including in the revised discussion, last paragraph on page 18. Regarding HDAC9, the overall levels of this transcript are very low and despite multiple attempts we have not been able to detect endogenous HDAC9 protein present in osteocytes. Therefore, the functional significance of the HDAC9 mRNA reductions seen here are currently unclear.

3. The N/C fractionation in Figure 5D is not as clean as in the other experiments. Is there a better experiment that could be shown? The authors should also find a way to normalize levels of Hdac4 and CRTC2 to a control protein.

We appreciate this point, and agree that the fractionation in Figure 5D wasn't as "clean" as other fractionations shown in this manuscript. For this reason, a revised version of Figure 5D is now shown from an independent experiment

comparing effects of PTH and SIK inhibitor on HDAC4 and CRCT2 subcellular localization. As is standard in the field, data in this figure are quantified as the percentage of total HDAC4 and CRCT2 protein present in the nucleus. A similar result is obtained if nuclear HDAC4 and CRCT2 levels are normalized to the loading control SP1, as shown below in Figure R1.

Figure R1. Top graphs are identical to the data from Figure 5D in the manuscript. Bottom graphs show the same densitometric quantification when the ratio of nuclear CRCT2 (left) or HDAC4 (right) is shown relative to the loading control SP1 instead. In either case, PTH and YKL-05-093 induce nuclear translocation of HDAC4 and CRCT2.

4. *The synthesis and early testing of YKL-05-093 is impressive. Supplementary Table 2 indicates that other kinases may also be affected. Because specificity is a potential issue, some in vivo and in vitro studies would be helpful. Does YKL-05-093 affect bone density or other bone properties in mice and bone formation in cell culture models?*

This is an extremely important point. As discussed in the first full paragraph on page 16, while YKL-05-093 is a potent SIK2 inhibitor with interesting effects *in vitro*, repeated dosing of mice with this compound proved to be toxic. Therefore, we turned our attention to a related small molecule SIK inhibitor YKL-05-099. The broad kinase profiling data of this inhibitor has been reported by Sundberg et al (2), and is overall quite similar to that of YKL-05-093. Importantly, YKL-05-099 has vastly improved pharmacokinetic and toxicologic profiles compared to YKL-05-093. As shown in Figure 8 of our revised manuscript, YKL-05-099 affects gene expression in osteocytes in much the way that YKL-05-093 does, in a SIK-dependent manner. More importantly, repeated *in vivo* dosing with YKL-05-099 increases osteoblast numbers, osteoblast activity, and trabecular bone mass.

5. *Figure 6 only shows data for a select group of genes. All the data should be shown, even negative data, in the supplemental figures.*

We appreciate the opportunity to respond to this important point. Revised Supplemental Figure 6E shows data from all the genes measured in this

analysis, categorized into genes regulated by YKL-05-093 that are completely SIK2/3-dependent, partially SIK2/3-dependent, and SIK2/3-independent. We believe that displaying all the data in this manner allows the reader to more broadly appreciate the on- and off-target effects of this compound in Ocy454 cells with respect to changes in gene expression.

6. The results in Figures 7A and 7B are very stunning due to the rapid disappearance of SOST. Do RANKL levels also change? In Figure 3C/D, were the IHC assays started within 90 minutes of PTH administration?

We agree that the changes in SOST mRNA and sclerostin protein observed *in vivo* in response to PTH or YKL-05-093 are rapid (within 90 minutes) and stunning. Indeed, rapid PTH-induced SOST/sclerostin down-regulation has been observed by many laboratories over the last 10 years (for example, (3).) Here we report that small molecule SIK inhibitors such as YKL-05-093 possess similar properties. RANKL mRNA levels rapidly increase in response to both PTH (Figure 3A) and YKL-05-093/YKL-05-099 (Figure 7B and new Figure 8C). Unfortunately, multiple attempts at immunohistochemistry to detect RANKL protein *in situ* with multiple commercially-available antibodies were unsuccessful in that unacceptable background/non-specific staining was present.

7. On page 7, the sentence mentioning the "+45kB downstream enhancer (figure 2G)" is confusing. Does that refer to the Sost or Mef2C enhancer?

We apologize for this confusion. Page 7 (last paragraph) now includes a clarification indicating that MEF2C binds to the SOST gene's +45 kB downstream enhancer.

8. Information on how phosphorylation affects Hdacs should be moved from the results section (page 6) to the introduction (page 4) for uninformed readers. On this same theme, the model in Figure 8 is incorrect if Hdac4/5 and CRTC2 are phosphorylated in the nucleus and then exported to the cytoplasm.

We appreciate this opportunity to clarify our manuscript. The first paragraph on page 4 of the introduction includes additional details about how dynamic phosphorylation/dephosphorylation affects class IIa HDAC subcellular localization. Given our findings that SIK2/3 are predominantly cytoplasmic, we believe that our model demonstrating cytoplasmic HDAC phosphorylation holds true.

Reviewer #2

Expert in SIKs

(Remarks to the Author):

In the submitted manuscript, the authors demonstrate the PTH-GS α -cAMP-HDAC 4/5-MEF2C-SOST axis in Ocy454 cells in vitro and then go on to show

that HDAC 4/5 is responsible for the PTH-induced down-regulation of SOST expression by administering PTH to HDAC 4/5 DKO mice *in vivo* (Fig. 3). The down-regulation was inhibited by the deletion of both HDAC4 and HDAC5 in mice.

Based on this result and the previously reported findings that class II HDACs are substrates of SIKs and that cAMP inhibits SIK2 through PKA-mediated phosphorylation in adipocytes and hepatocytes, the authors hypothesized that SIKs mediate PTH function in osteocytes. Supporting this hypothesis, the authors demonstrate that SIK2 mediates the PTH-induced down-regulation of SOST expression in Ocy454 cells. In addition, the authors found that SIK2 also mediates the PTH-induced up-regulation of RANKL expression in Ocy454 cells. It has been known that CREB activates RANKL transcription and that SIKs inhibit CREB function by phosphorylating CRTC, a co-activator of CREB. The authors demonstrate that PTH promotes the translocation of CRTC2 to the nucleus and that CRTC2 is responsible for the PTH-induced up-regulation of RANKL expression.

Finally, the authors investigate the role of SIKs on the PTH regulation of SOST and RANKL mRNA expressions in mice. For this purpose, they identified two small molecules, YKL-04-114 and YKL-05-093, that inhibit SIK by dephosphorylating pS358 (Fig. 5J). YKL regulated the SOST and RANKL mRNA expressions in a SIKs- and CRTC2-dependent manner, respectively, in Ocy454 cells. RNA seq analysis showed that YKL mimics PTH. Finally, treatment of mice with YKL decreased SOST expression and increased RANKL expression in osteocytes.

This work has the potential to significantly advance our understanding of PTH action on osteocytes, which has important implications biologically and medically. Of special note is the elegant work that shows SIKs are substantial mediators of PTH. Another strong point of this study is the identification of YKLs as inhibitors of SIKs and mimics of PTH action in osteocytes. While the findings of the paper are significant advance, several points need to be clarified.

Major

1. The pathway analyses were initially done in Ocy454 cells *in vitro*. The authors then confirmed the function of HDAC 4/5 in the PTH-induced down-regulation of SOST in mice with HDAC 4/5 DKO background *in vivo*. Further experiments showed that SIKs are mediators of PTH in Ocy454 cells *in vitro* and in mice treated with YKL *in vivo*. As shown by the authors, YKL also affected other enzymes in addition to SIKs. Although the authors show YKL action in Ocy454 cells via SIKs *in vitro*, it is not known whether this mechanism is also the case *in vivo*. Experiments such as treating SIK KO or cKO mice with PTH would confirm the authors' conclusion.

We completely agree that additional *in vivo* experiments are needed to confirm a role for SIK2 in mediating PTH effects, especially given the promiscuous nature of the YKL-05-093 inhibitor. For this reason, we have obtained SIK2 conditional knockout mice and bred them to the DMP1-Cre delete strain which is active in

mature osteoblasts and osteocytes. In revised Figure 4I-K, we characterize responses to acute PTH administration in these animals *in vivo*. Similar to what was observed in SIK2 deficient Ocy454 cells, SIK2 knockout in osteocytes *in vivo* blocks the ability of PTH to regulate RANKL and SOST, while CITED1 up-regulation remains intact. Therefore, we believe that this additional experiment significantly strengthens our model that SIK2 is required for acute responses to PTH both *in vitro* and *in vivo*.

2. This reviewer would expect SIK inhibitor compounds, such as YKL, could cure osteoporosis. Such a finding would increase the significance of this study. What is the bone phenotype of mice treated with YKL for certain period of time? Does it mimic the phenotype of mice treated with PTH? It is known that animals treated with PTH show specific histologies in the cortex and spongiosa of bone.

This is an excellent and extremely important point. While YKL-05-093 caused significant toxicity with repeated dosing, YKL-05-099 is another potent SIK inhibitor suitable for *in vivo* use. As shown in Figure 8 of our revised manuscript, YKL-05-099 affects gene expression in osteocytes similar to YKL-05-093 in a SIK-dependent manner. More importantly, repeated *in vivo* dosing with YKL-05-099 increases osteoblast numbers, osteoblast activity, and trabecular bone mass.

Specific

1. Fig. 2D. Should HDAC4 pS259 be HDAC5 pS259? Why did the amount of HDAC5 pS259 increase with the addition of PTH or Forskolin in clone #11?

We appreciate the opportunity to fix this figure labeling error. Overall, while PTH and SIK inhibitors reliably and reproducibly reduce levels of HDAC4 S246 phosphorylation, data regarding HDAC5 S259 phosphorylation are more variable. We believe that this may be due to the fact that HDAC7 has a very similar molecular weight to HDAC5, and this phospho-specific antibody also recognizes HDAC7 S155. Therefore, variable levels of HDAC7 phosphorylation in these experiments may interfere with our ability to reliably detect HDAC5 N-terminal phosphorylation using this reagent.

2. Sup. Fig. 2E. The amount of HDAC5 pS259 did not decrease with the addition of PTH or FSK.

Please see above discussion regarding the potential issues with using this antibody to confidently study HDAC5 S259 phosphorylation.

3. Both shSIK2 and shSIK3 were necessary for substantial dephosphorylation of HDAC 4/5 (Fig. 4A), but PTH did not induce phosphorylation of SIK3 (Fig. 4B). In addition, shSIK2 suppressed both the PTH-induced dephosphorylation of HDAC 4/5 (Fig. 4E) and the PTH-induced down-regulation of SOST expression (Fig.

4F). Does SIK3 also play a role in the changed expression levels of pHDAC 4/5, SOST, and RANKL that are regulated by PTH?

We appreciate the opportunity to clarify this important issue. As noted by the reviewer, PTH does not promote SIK3 phosphorylation using a phospho-specific antibody designed against a PKA site (T469) or a consensus PKA substrate antibody. Revised Figure 4D, F, and G show gene expression studies in Ocy454 cells lacking SIK2 or SIK3 in response to PTH. While SIK3 deficiency has no appreciable effects on acute responses to PTH, SIK2 is required for PTH to regulate SOST and RANKL expression in these short-term experiments. The reviewer astutely points out that basal HDAC4 phosphorylation is only reduced by the combination of SIK2 and SIK3 knockdown. We believe that this reflects different requirements for kinases that regulate basal versus PTH-stimulated HDAC4 phosphorylation.

4. Did the authors try to examine the effects of inactivated SIKs or the addition of YK-04-114 on the phosphorylation levels of CRTCs? Are they like the effects seen on HDACs?

PTH and SIK inhibitors clearly promote the nuclear translocation of CRTC2 (Figure 5D) and its binding to the RANKL enhancer (Figure 4M). Unfortunately, there are no commercially-available phospho-specific CRTC2 antibodies suitable to detect changes in endogenous CRTC2 phosphorylation. We did purchase a reagent from Bioss (bs-3415) that reportedly detects CRTC2 phospho-S177. However, despite multiple attempts using this antibody for immunoblotting on lysates or CRTC2 immunoprecipitates, no reliable signal was detected.

5. *shSIK2* was sufficient to suppress the PTH effects on SOST (Fig. 4F) and RANKL (Fig. 4G). Both *shSIK2* and *shSIK3*, however, were needed to suppress the YKL-induced down-regulation of SOST transcription (Figs. 5F, G). Please explain the discrepancy between these two results. In other words, what role does SIK3 play? In addition, how about regulation of RANKL?

This is a very important point, and we appreciate the opportunity to clarify this. SIK2 deficiency (*in vitro* and *in vivo*) is sufficient to abrogate PTH-induced RANKL and SOST gene regulation. PTH signaling leads to SIK2 phosphorylation at multiple sites (S358, S343, T484) whose phosphorylation is known to potently inhibit SIK2 cellular activity. In contrast, YKL-05-093 and YKL-05-099 function as canonical SIK kinase inhibitors, and therefore block ATP binding to the active site of targeted kinases. These YKL series compounds target both SIK2 and SIK3 by this mechanism. Furthermore, both SIK2 and SIK3 contribute to basal HDAC4/5 phosphorylation (Figure 4A); therefore, we believe that our findings that combined SIK2/3 deficiency is required to lose many cellular effects of YKL-05-093 and YKL-05-099 is internally consistent. We also appreciate the opportunity to clarify the issue of RANKL gene regulation. Figure 8B of the revised manuscript shows that both YKL-05-093 and YKL-05-099 potently up-regulate

RANKL in control, but not SIK2/3-deficient, Ocy454 cells. Therefore, RANKL up-regulation by YKL series compounds is also an 'on-target' (SIK dependent phenomenon).

6. *Fig. 5A graph. Please explain the X- and Y-axes.*

We have revised the figure legend to fully explain this graph. The K_d of YKL-05-093 for recombinant SIK2 was determined using methods fully described in the manuscript's methods section. The y-axis shows bound SIK2 and the x-axis shows increasing doses of YKL-05-093.

7. *How does YKL inhibit SIKs? The authors show it dephosphorylates SIK2 pS358 (Fig. 5J). Does YKL act on SIKs directly or does it act on upstream molecules that ultimately phosphorylate SIK2 S358?*

We appreciate the opportunity to clarify this point. As described above, PTH promotes SIK2 S358 phosphorylation which inhibits its cellular activity. YKL-05-093 and YKL-05-099 inhibit SIK2 by functioning as canonical kinase inhibitors, in a manner similar to that of the parental HG-9-91-01 inhibitor (2, 4, 5).

8. *Please describe the route by which mice were treated with sclerostin antibody, PTH and YKL. Subcutaneous injection or oral administration?*

We appreciate the opportunity to provide this important information. Page 24-25 of the revised methods sections states that sclerostin antibody and PTH were given via subcutaneous administration, and YKL-05-093 and YKL-05-099 were given via intraperitoneal administration.

9. *Fig. 7C. For the immunohistochemical analysis, Please indicate how many hours after treatment with YKL the sclerostin was examined.*

The figure legend for Figure 7C has been clarified to include this information: as in the RNA analysis, sclerostin immunostains were performed 2 hours after IP treatment with either vehicle or YKL-05-093.

10. *What are the bone phenotypes of mice lacking SIKs? Have they been reported?*

A paragraph in the discussion (page 21-22) has been added to review the skeletal and extra-skeletal phenotypes of SIK2 and SIK3 mutant mice. As described above, our revised manuscript now contains *in vivo* data demonstrating a requirement for SIK2 in acute responses to PTH at the level of RANKL and SOST gene expression.

11. *Discussion, first paragraph, last sentence. "Whether this pathway operates in other extraskeletal PTH/PTHrP target cells, such as renal epithelial cells (45), T*

lymphocytes (46), and adipocytes (47) remains to be determined." Chondrocytes should be included in the list of cells.

We regret this glaring omission, especially since a major focus of our laboratory over the past 2 decades has been how PTHrP controls chondrocyte biology. Chondrocytes have been added to this list of cells with an appropriate reference.

12. Discussion, page 16, second paragraph. "While both pathways utilize a class IIa HDAC/MEF2 mechanism of action..." A class IIa HDAC/MEF2 mechanism should be viewed as a SIKs/a class IIa HDAC/MEF2 mechanism and discussed in this respect, because chondrocytes have been reported to use SIK3 (Sasagawa et al., Development 139, 1153-1163, 2012).

This is an excellent point. Until now, potential links between PTHrP signaling and the fascinating SIK3 knockout phenotype have not been made. We hope that our current manuscript on PTH signaling through SIKs in osteocytes prompts future research into whether SIKs participate in PTHrP signaling in chondrocytes. This paragraph of the discussion has been revised accordingly.

Reviewer #3

Expert in bone remodelling, Wnt signalling

(Remarks to the Author):

Wein et al studied the mechanism through which PTH suppresses Sost and stimulates rankl transcription in osteocytes. They have provided evidence that PTH inhibits SIK2 activity to promote nuclear entry of Hdac4/5 in the case of Sost suppression and that of CRTC2 in the case of Rankl induction. Furthermore, the authors developed novel SIK inhibitors to demonstrate the role of SIK in the regulation of Sost and Rankl. Overall, the biochemical evidence for such regulation is strong. However, the overall contribution of such a mechanism to the bone anabolic effect of PTH needs to be clarified to strengthen the paper. Detailed comments are as follows.

1) The authors open the paper with supplemental figure1 to argue for the functional interaction of Hdac4/5 and Sost. However, because hdac5+/- does not have a bone phenotype, Fig. S1A does not support that hdac5 works through Sost but rather that the Sost+/- phenotype requires intact hdac5. In Fig. S1D, the authors should use anova to assess statistical significance of the interaction between anti-Sost and Hdac5 deletion. Finally, do the authors have evidence that anti-Sost rescues the bone phenotype in Hdac4/5 double mutant?

These are all excellent points. We completely agree that the most straightforward conclusion from the genetic interaction study in Figure S1A is that the SOST+/- phenotype requires both alleles of HDAC5 to be present. As such, we have slightly revised the text describing this experiment (page 5, first paragraph under Results) to state that we initially sought to determine whether HDAC5 and SOST interact *in vivo* to control bone mass.

We consulted with a biostatistician regarding the point concerning statistical analysis of the experiment performed in Figure S1D. As suggested, 2-way ANOVA was first determined which revealed that there was a significant interaction ($p < 0.01$) between HDAC4/5 deficiency and Scl-Ab treatment. At this point, we performed posthoc t tests to evaluate within-group differences. These values are shown on the graph in Figure S1D.

Revised Figure S1D also shows the results we have recently obtained when HDAC4/5 DKO mice were treated with Scl-Ab. Surprisingly, Scl-Ab did not boost bone mass in these animals. Several interesting phenotypes are present in the HDAC4/5 DKO mice, as described in the text on page 6 (first paragraph). As detailed in the revised discussion (page 19, second paragraph), increased sclerostin levels cannot fully explain the phenotype of these DKO mice. We believe that the major point of this current manuscript is that a major arm of PTH signaling involves SIK inhibition. A complete understanding of the skeletal phenotype of the HDAC4/5 DKO mice is beyond the scope of the current study.

2) Have the authors tested potential anabolic effect of PTH in Hdac4/5 mutants?

As shown in Figure 6, PTH and YKL-05-093 regulate the expression of hundreds of genes in osteocytes. In addition to driving class IIa HDACs into the nucleus, PTH and YKL-05-093 promote nuclear translocation of CRTC2. Therefore, although class IIa HDACs are required for acute PTH-induced SOST suppression, we did not expect to see a blunted anabolic effect of intermittent PTH treatment in HDAC4/5 DKO mice. New Figure S4A shows that intermittent PTH increases cancellous bone mass in all HDAC mutant strains tested. This finding leads us to conclude that HDAC-independent pathways contribute to PTH-induced bone anabolism (revised manuscript, page 8, last paragraph). Given the fact that *in vivo* YKL-05-099 SIK inhibitor treatment increases bone mass and osteoblastic bone formation in a manner similar to that of PTH, we believe that SIK-dependent but HDAC4/5-independent signaling pathways are likely responsible.

3) Does the SIK inhibitor have a bone anabolic effect?

Yes - this is an extremely important point. While YKL-05-093 caused significant toxicity with repeated dosing, YKL-05-099 is another potent SIK inhibitor suitable for prolonged *in vivo* use. As shown in Figure 8 of our revised manuscript, YKL-05-099 affects gene expression in osteocytes similar to YKL-05-093 in a SIK-dependent manner. More importantly, repeated *in vivo* dosing with YKL-05-099 increases osteoblast numbers, osteoblast activity, and trabecular bone mass.

4) Fig. S4E showed great decrease in cAMP induction by PTH in SIK2 knockdown cells. This result weakens the argument that the loss of Sost and Rankl response to PTH was strictly due to SIK2 deficiency. Did the authors test

the effect of forskolin on Sost/Rankl expression in these cells?

This is an important and very thoughtful observation. SIK2 deficient Ocy454 cells show reduced cAMP-up-regulation in response to parathyroid hormone. We appreciate the excellent suggestion to test the effects of forskolin in these cells. Revised Figure 4H shows this data. While PTH-induced cAMP upregulation is blunted in SIK2 deficient cells, forskolin treatment robustly increases cAMP levels in control and shSIK2 cells. Despite large forskolin-induced increases in cAMP in SIK2-deficient cells, forskolin still does not regulate RANKL and SOST in these cells. Therefore, these data further support our model in which cAMP signals to inactivate SIK2. Without SIK2 present, this cellular switch is absent and therefore, CRT2 and class IIa HDAC target genes cannot be regulated.

Reviewer #4

Expert in organic chemistry synthesis

This reviewer has been asked to focus on the chemical synthesis aspects of the paper, so all comments are limited to this area of the work only.

As part of this manuscript the authors report the synthesis of YKL-05-093 and YKL-04-114. The procedures are sufficiently detailed to allow for a peer to replicate the work described. However the analytical information reported for each compound is insufficient to confirm both the identity and/or purity of the compounds (in line with normal organic synthesis practice). This must be rectified before this article can be considered for publication.

For compound 2: This is a known compound and reaction (Brossmer,R.; Roehm,E. Justus Liebigs Annalen der Chemie, 1966 , vol. 692 , p. 119 - 133, along with numerous modern patent references). References must be provided for this compound.

As a known compound the minimum acceptable analytical data to be reported is: 1H NMR, 13C NMR, IR, Low res MS (given in this case) and melting point (referenced against that known, in this case 52-53 C)

For compounds 3,4,5, YKL-05-093 and YKL-04-114 as unknown compounds the minimum acceptable analytical data to be reported is: 1H NMR, 13C NMR, IR, Low res MS, melting point, High res MS OR elemental analysis (to confirm formula). Purity needs to be demonstrated, this is normally done through inclusion of the 1H and 13C NMR spectra in the supplemental information (additional data such as LC maybe included).

We simplified the synthetic scheme based on the reviewer's suggestion. The compounds 2,3,4,5 in the initial manuscript have been named as compounds 1,2,3,4 in the current manuscript.

Compound 1 was prepared according to recent patent literature (reference provided), and the characterization matches literature precedence.

Analytical data of ^1H NMR, ^{13}C NMR, and HRMS are provided for new compounds 2,3,4, YKL-04-114, and YKL-05-093 based on the chemical identity requirements listed on the Nature Communication website. ^1H NMR and ^{13}C NMR spectra are also provided in new Supplemental Figures 8.

Additional points;

1) For compound 5 where ^1H NMR is given. The authors report the benzylic methylene as 4.74 (dd, $J = 5.5, 1.6$ Hz, 2H), which looks odd. Can the authors confirm this is not either a mixture of rotamers or a miss-assigned geminally coupled system.

These two protons are two geminally coupled protons. They were mistakenly assigned in the initial manuscript. The assignments were corrected in the current manuscript. As mentioned above, compound 5 in the initial manuscript have been named as compound 4 in current manuscript.

2) For compounds YKL-05-093 and YKL-04-114 the geminal coupling for the methylene in the dihydropyrimidinone ring suggests that these protons are in very different environments. Considering the structures of YKL-05-093 and YKL-04-114 have the authors considered that there may be restricted rotation around the C-N bond between the dihydropyrimidinone ring and the right hand dimethylresorcinol? This molecule may exist as a racemic mixture of atropisomers (this could be examined by chiral HPLC or possibly x-ray crystallography). This should be tested as it is possible that one enantiomer could have improved biological activity.

It's possible that atropisomers exist for these two compounds, due to the potential restricted rotation of the C-N bond. All attempts to crystallize both compounds failed. Chiral HPLC was attempted for both compounds using several conditions, but no sign of atropisomers were observed. This suggests that even if atropisomers exist in these molecules, the rotation energy barrier of the C-N bond could possibly be lower than 20 kcal/mol or fall into the region that's between 20-30 kcal/mol, which would make the separation of these atropisomers nearly impossible (6). As a result, these compounds were developed as either a single compound or as an unseparated mixture for SIK inhibition.

Sincerely,

Marc N. Wein

Henry M. Kronenberg

REFERENCES

1. Shimizu E, Selvamurugan N, Westendorf JJ, Olson EN, Partridge NC. HDAC4 represses matrix metalloproteinase-13 transcription in osteoblastic cells, and parathyroid hormone controls this repression. *The Journal of biological chemistry*. 2010 Mar 26;285(13):9616-26. PubMed PMID: 20097749. Pubmed Central PMCID: 2843211.
2. Sundberg TB, Liang Y, Wu H, Choi HG, Kim ND, Sim T, et al. Development of Chemical Probes for Investigation of Salt-Inducible Kinase Function in Vivo. *ACS chemical biology*. 2016 Jun 6. PubMed PMID: 27224444.
3. Keller H, Kneissel M. SOST is a target gene for PTH in bone. *Bone*. 2005 Aug;37(2):148-58. PubMed PMID: 15946907.
4. Clark K, MacKenzie KF, Petkevicius K, Kristariyanto Y, Zhang J, Choi HG, et al. Phosphorylation of CRTCL3 by the salt-inducible kinases controls the interconversion of classically activated and regulatory macrophages. *Proceedings of the National Academy of Sciences of the United States of America*. 2012 Oct 16;109(42):16986-91. PubMed PMID: 23033494. Pubmed Central PMCID: 3479463.
5. Sundberg TB, Choi HG, Song JH, Russell CN, Hussain MM, Graham DB, et al. Small-molecule screening identifies inhibition of salt-inducible kinases as a therapeutic strategy to enhance immunoregulatory functions of dendritic cells. *Proceedings of the National Academy of Sciences of the United States of America*. 2014 Aug 26;111(34):12468-73. PubMed PMID: 25114223. Pubmed Central PMCID: 4151730.
6. LaPlante SR, Edwards PJ, Fader LD, Jakalian A, Hucke O. Revealing atropisomer axial chirality in drug discovery. *ChemMedChem*. 2011 Mar 7;6(3):505-13. PubMed PMID: 21360821.

REVIEWERS' COMMENTS:

Reviewer #1 (Remarks to the Author):

Thank you for addressing my previous queries in a thoughtful way. I have just 2 remaining and relatively minor comments.

1. Please indicate the sex of the animals used for experiments in the legends for Figures 1, 3, and 7.
2. Figure 9 is still confusing because it does not indicate how pHdac4/5 becomes Hdac4/5 before (or after) they enter the nucleus. I would recommend adding 14-3-3 proteins and a phosphatase to the figure to illustrate a more complete picture of the molecular pathway.

Excellent work! Congratulations!

Reviewer #2 (Remarks to the Author):

The authors have adequately responded to this reviewer's queries.

Minor comments.

1. Line 221 on page 9, should Figure S4E be S4F?
2. Line 420 on page 18, the word "extrasketal" should be changed, because chondrocytes are skeletal cells.
3. Line 525 on page 23, please check grammar of the sentence in. "This is consistent was potential redundancy between two kinases, ...".

Reviewer #3 (Remarks to the Author):

The authors have addressed all of my concerns.

Reviewer #4 (Remarks to the Author):

The author has suitably addressed my previous comments. The inclusion of spectral data for the compounds synthesized is of sufficient quality to allow purity to be established.

Please can the authors include a footnote or comment in the ESI on their investigations into atropisomerism and that no enantiomers could be resolved, suggesting a low barrier to inter-conversion. Happy to approve the paper for publication.

We found the reviewers' final comments to be extremely useful, responses to each point raised are found below, with our responses underlined and italicized.

Reviewer #1 (Remarks to the Author):

Thank you for addressing my previous queries in a thoughtful way. I have just 2 remaining and relatively minor comments.

1. Please indicate the sex of the animals used for experiments in the legends for Figures 1, 3, and 7. *We appreciate the opportunity to clarify this important point. The figure legends for these figures have been revised to indicate that male mice were used for these experiments.*

2. Figure 9 is still confusing because it does not indicate how pHDac4/5 becomes Hdac4/5 before (or after) they enter the nucleus. I would recommend adding 14-3-3 proteins and a phosphatase to the figure to illustrate a more complete picture of the molecular pathway.

This is an excellent point. Previous literature supports that phosphorylated HDAC4/5 are retained in the cytoplasm through binding to 14-3-3 proteins. These are now included in the final version of Figure 9. In addition, a phosphatase must be acting to dephosphorylate HDAC4/5 and CRTC2 when SIK activity is reduced. Our data indicates that PP2A is not the sole phosphatase that accomplishes this function. Future studies will be needed to identify and characterize this phosphatase, as we point out in the revised figure legend.

Excellent work! Congratulations!

Reviewer #2 (Remarks to the Author):

The authors have adequately responded to this reviewer's queries.

Minor comments.

1. Line 221 on page 9, should Figure S4E be S4F?

Yes, this typographical error has been corrected

2. Line 420 on page 18, the word "extraskkeletal" should be changed, because chondrocytes are skeletal cells.

Of course, the word "extraskkeletal" has been replaced by the word "other".

3. Line 525 on page 23, please check grammar of the sentence in. "This is consistent was potential redundancy between two kinases, ...".

This grammatical error was fixed – the word “was” has been replaced by the word “with”.

Reviewer #3 (Remarks to the Author):

The authors have addressed all of my concerns.

Reviewer #4 (Remarks to the Author):

The author has suitably addressed my previous comments. The inclusion of spectral data for the compounds synthesized is of sufficient quality to allow purity to be established.

Please can the authors include a footnote or comment in the ESI on their investigations into atropisomerism and that no enantiomers could be resolved, suggesting a low barrier to inter-conversion.

Happy to approve the paper for publication.

As requested, a statement about our investigations into atropisomerism has been included in the supplementary information detailing synthetic methods.